# Interaction mapping of endoplasmic reticulum ubiquitin ligases identifies modulators of innate immune signalling

**Emma J Fenech[1†], Federica Lari[1], Philip D Charles[2], Roman Fischer[2], Marie Laétitia-Thézénas[2], Katrin Bagola[1‡], Adrienne W Paton[3], James C Paton[3], Mads Gyrd-Hansen[1], Benedikt M Kessler[2,4], John C Christianson[1,5,6]\***

[1]Ludwig Institute for Cancer Research, Nuffield Department of Medicine, University of Oxford, Oxford, United Kingdom; [2]TDI Mass Spectrometry Laboratory, Target Discovery Institute, University of Oxford, Oxford, United Kingdom; [3]Research Centre for Infectious Diseases, Department of Molecular and Biomedical Science, University of Adelaide, Adelaide, Australia; [4]Chinese Academy of Medical Sciences Oxford Institute, Nuffield Department of Medicine, University of Oxford, Oxford, United Kingdom; [5]Nuffield Department of Orthopaedics, Rheumatology, and Musculoskeletal Sciences, University of Oxford, Botnar Research Centre, Oxford, United Kingdom; [6]Oxford Centre for Translational Myeloma Research, University of Oxford, Botnar Research Centre, Oxford, United Kingdom

**\*For correspondence:**
john.christianson@ndorms.ox.ac.uk

**Present address:** [†]Department of Molecular Genetics, Weizmann Institute of Science, Rehovot, Israel; [‡]Paul Ehrlich Institut, Federal Institute for Vaccines and Biomedicine, Langen, Germany

**Competing interests:** The authors declare that no competing interests exist.

**Abstract** Ubiquitin ligases (E3s) embedded in the endoplasmic reticulum (ER) membrane regulate essential cellular activities including protein quality control, calcium flux, and sterol homeostasis. At least 25 different, transmembrane domain (TMD)-containing E3s are predicted to be ER-localised, but for most their organisation and cellular roles remain poorly defined. Using a comparative proteomic workflow, we mapped over 450 protein-protein interactions for 21 stably expressed, full-length E3s. Bioinformatic analysis linked ER-E3s and their interactors to multiple homeostatic, regulatory, and metabolic pathways. Among these were four membrane-embedded interactors of RNF26, a polytopic E3 whose abundance is auto-regulated by ubiquitin-proteasome dependent degradation. RNF26 co-assembles with TMEM43, ENDOD1, TMEM33 and TMED1 to form a complex capable of modulating innate immune signalling through the cGAS-STING pathway. This RNF26 complex represents a new modulatory axis of STING and innate immune signalling at the ER membrane. Collectively, these data reveal the broad scope of regulation and differential functionalities mediated by ER-E3s for both membrane-tethered and cytoplasmic processes.

## Introduction

The endoplasmic reticulum (ER) is the largest membrane-bound organelle in eukaryotic cells, comprised of a complex network of sheets, tubules, junctions and contact sites that can occupy more than 35% of the entire cell volume (*Valm et al., 2017*) and a significant fraction of the total membrane surface area. The continuous lattice it forms with the nuclear envelope (NE) makes extensive and dynamic contacts through distal projections with mitochondria (*Kornmann et al., 2009*), peroxisomes (*Smith and Aitchison, 2013*), endosomes (*van der Kant and Neefjes, 2014*; *Hönscher and Ungermann, 2014*), plasma membrane (*Saheki and De Camilli, 2017*) and lipid droplets (reviewed in *Wu et al., 2018a*). Within this extensive network, the ER accommodates biogenic, metabolic and regulatory multi-subunit transmembrane domain (TMD)-containing protein complexes that span the lipid bilayer and simultaneously carry out processes essential for cellular homeostasis.

Post-translational modification by ubiquitin (Ub) targets proteins for degradation, promotes interactions, directs subcellular localisation, or drives signalling (*Komander and Rape, 2012*). The enzymatic cascade conjugating and extending Ub chains on proteins throughout the mammalian cell uses one (or more) of the >600 Ub ligases (E3s) to provide reaction specificity by bringing substrates and Ub conjugating enzymes (E2s) in proximity (*Li et al., 2008*). E3s distinguish substrates either directly through dedicated binding domains/surfaces, or indirectly by assembling co-factors into specialised multi-subunit complexes with recognition and recruitment capabilities (reviewed in *Zheng and Shabek, 2017*). Within aqueous environments of the cytoplasm and nucleus, freely diffusing E3s access substrates with reduced spatial impediment. In contrast, E3s embedded within lipid bilayers by virtue of one or more TMDs or lipid anchor, have pre-determined orientation and lateral motion restricted to the planar membrane where they reside. Eukaryotic membrane-embedded E3s are found in the ER (*Claessen et al., 2012*), inner nuclear membrane (INM) (*Foresti et al., 2014*), mitochondria (*Li et al., 2008*), Golgi (*Stewart et al., 2011*), endosomes and plasma membrane (*Piper and Lehner, 2011*). The RING and HECT domains of E3s coordinating Ub transfer are exclusively exposed to the cytoplasm (and nucleus), enabling the access of cytosolic, nuclear, and proximal membrane proteins. Moreover, lumenal proteins from within the ER also reach the RING domain of at least one membrane-bound E3 (Hrd1), by way of an aqueous channel it forms within the lipid bilayer (*Schoebel et al., 2017*). Thus, membrane bound E3s arguably serve as broad platforms for ubiquitination within the cell. Understanding how E3 complexes recognise substrates in and around membranes, how they can coordinate efficient Ub conjugation, how they regulate access to them, and which cellular processes they regulate, are important biological questions that remain outstanding.

ER-associated degradation (ERAD) has been the principal modality for understanding E3 function in the ER. Secretory cargo transiently or terminally misfolded during biogenesis in the ER is prone to aggregation that can cause proteotoxic stress, necessitating removal from the organelle that ultimately ends in its degradation (*Vembar and Brodsky, 2008*; *Claessen et al., 2012*; *Christianson and Ye, 2014*). ERAD serves in tandem with chaperone-mediated folding and assembly processes, acting as an integral facet of the organelle's quality control machinery (*Hebert and Molinari, 2007*). How we envisage E3 function at the ER has been shaped by extensive studies on the evolutionarily conserved Hrd1 (*Hampton et al., 1996*; *Carvalho et al., 2006*; *Carvalho et al., 2010*; *Christianson et al., 2008*; *Bays et al., 2001*; *Christianson et al., 2012*; *Stein et al., 2014*; *Hwang et al., 2017*). With as many as eight TMDs (*Schoebel et al., 2017*), a cytoplasmic RING domain and an extended C-terminus with low complexity (*Schulz et al., 2017*), Hrd1 scaffolds specialised lumenal, integral membrane, and cytosolic co-factors such as the AAA ATPase VCP/p97 (*Ye et al., 2001*; *Ye et al., 2003*) to form multi-component complexes that coordinate recognition, retrotranslocation, and ubiquitination of misfolded secretory cargo (*Carvalho et al., 2006*; *Christianson et al., 2012*).

E3 complexes not only control the quality of secretory cargo but also adjust the abundance (and hence activity) of ER-resident membrane proteins through 'regulatory ERAD' (*Ruggiano et al., 2014*). As the primary site for phospholipid and sterol biosynthesis (*Schwarz and Blower, 2016*), degradation of ER-resident rate-limiting enzymes such as 3-hydroxy-3-methyl-glutaryl-coenzyme A reductase (HMGCR) by gp78/AMFR (*Song et al., 2005*), RNF139/Trc8 (*Jo et al., 2011a*) and RNF145 (*Jiang et al., 2018*; *Menzies et al., 2018*), and squalene monooxygenase (SM) by MARCH6/Doa10 (*Foresti et al., 2013*; *Zelcer et al., 2014*), help tune the output of this pathway to maintain homeostasis. Both gp78 and RNF145 use Insig1/2 as adaptors to recruit and degrade HMGCR (*Song et al., 2005*; *Jiang et al., 2018*). Other processes at the ER membrane regulated by E3s and their interactors/adaptors include calcium flux (*Lu et al., 2011*), innate immune signalling (*Ishikawa and Barber, 2008*), antigen presentation (reviewed in *van den Boomen and Lehner, 2015*), endosome diffusion (*Jongsma et al., 2016*), ER morphology (*Zhao et al., 2016*) and apoptotic signalling (*Wu et al., 2018b*).

E3 complexes represent important post-translational regulatory modules tending to the protein landscape of the mammalian ER, but most have not been extensively characterised. We developed a comparative proteomic workflow to define interactions of ER-resident E3s that sought to identify networks involved in maintaining ER and/or cellular homeostasis. Among the E3 interactors confidently identified were proteins previously reported in connection with lipid regulation, calcium flux, quality control, as well as new interactors involved in innate immune signalling.

# Results

## Isolation and discovery of ER-resident E3 interactors

Of the >600 E3s present in the human proteome, about 10% contain TMDs that tether affiliated processes to lipid bilayers. Starting from previous reports (*Neutzner et al., 2011*; *Maruyama et al., 2008*; *van de Weijer et al., 2014*) and topology predictions, we shortlisted 25 E3s demonstrated to reside in the ER membrane. Selected ER-resident E3s (ER-E3s) are topologically and structurally disparate, arise from different phylogenetic lineages, but commonly possess one or more TMDs and a cytosolic RING-domain (*Figure 1A*). With a design to determine cognate ER-E3 co-factors and substrates, we developed a generalised expression protocol coupled to a comparative immunoprecipitation liquid chromatography tandem mass spectrometry (IP-LC-MS/MS) workflow that enriched for proteins interacting at the ER. We generated 25 stable, individual, HEK293 cell lines, each expressing a FLAG-HA (FH)-tagged E3 inducible by doxycycline (DOX) using Flp-In recombination (see Materials and methods, *Figure 1B*). Positioning of the FH-tag at either E3 terminus was determined

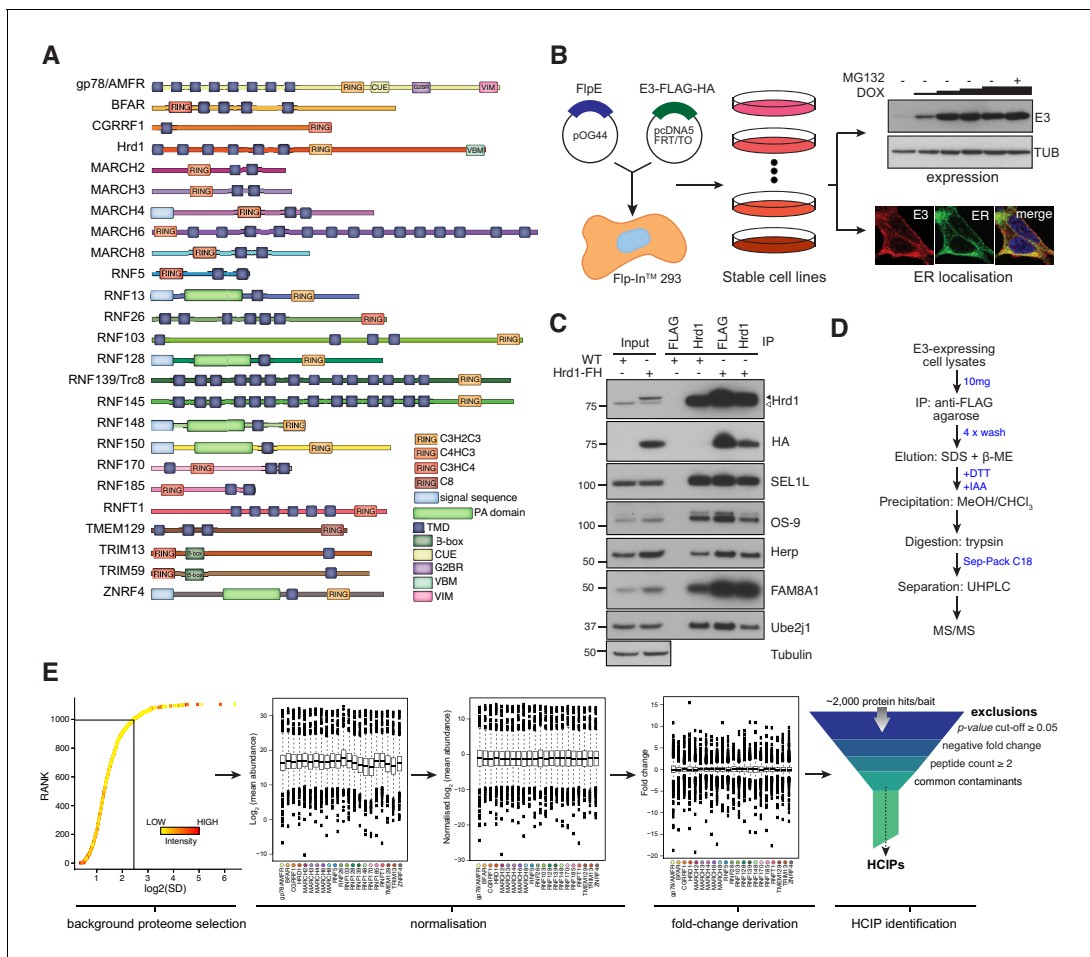

**Figure 1.** Proteomic analysis of ER-resident ubiquitin ligases. (**A**) ER-resident E3s and their predicted domains. (**B**) Workflow to generate and validate Flp-In293 cell lines stably expressing FLAG-HA-tagged E3s (FH-E3 or E3-FH). Each Flp-In293 cell line stably integrating a tagged E3 was screened for induction and expression over increasing concentrations of DOX and MG132 treatment by western blot (anti-FLAG) as well residency in the ER by immunofluorescence, evaluating colocalisation with markers of the ER, calnexin or KDEL. (**C**) Co-immunoprecipitation profiles of endogenous Hrd1 and DOX-induced Hrd1-FH prepared in 1% LMNG and isolated by anti-Hrd1 or anti-FLAG, as indicated. Input (20% of total IP) is also shown. (**D**) Workflow of sample preparation for LC-MS/MS analysis. (**E**) Bioinformatic processing pipeline for identification of high-confidence candidate interacting proteins (HCIPs) for the E3 baits.

The online version of this article includes the following figure supplement(s) for figure 1:

**Figure supplement 1.** Expression and localisation of FLAG-HA-E3s.

by taking into account prior experimental observations (*Neutzner et al., 2011*), predicted topology, and proximity to RING domains or other prominent structural features (*Figure 1A*, *Supplementary file 1* - Supplemental Table 1). Variable induction parameters sought to produce comparable E3 expression levels from each cell line (*Figure 1—figure supplement 1A*). More than 80% of ER-E3s tested (21/25) could be detected from whole cell lysates (WCL), accumulating in a DOX-dependent manner that reflects relative stability (*Figure 1—figure supplement 1A*). Including MG132 along with DOX enhanced detection of some E3s (e.g. RNF26, BFAR, TMEM129), indicative of intrinsic instability and constitutive turnover by proteasome-mediated degradation. E3s co-localised exclusively (or partly) with the ER-markers calnexin or KDEL (*Figure 1—figure supplement 1B*), consistent with their expected residency in the ER.

To assess whether tagged ER-E3s faithfully reproduce endogenous complexes, we compared the interaction profile of Hrd1 (*Christianson et al., 2012*; *Lilley and Ploegh, 2005*; *Mueller et al., 2008*; *Hosokawa et al., 2008*; *Schulz et al., 2017*; *Hwang et al., 2017*) with that of DOX-induced Hrd1-FH. Cofactors including SEL1L, FAM8A1, OS-9, Herp, and UBE2J1 were comparably co-immunoprecipitated (co-IPed) by both Hrd1 and Hrd1-FH (*Figure 1C*). Moreover, velocity sedimentation revealed that Hrd1-FH complexes migrate in fractions that overlap with those formed by endogenous Hrd1 (*Figure 1—figure supplement 1C*). From this, we anticipate that bona fide protein-protein interactions of candidate E3s will be recapitulated by the exogenously expressed E3s.

Preserving native interactions between TMD-containing E3s and co-factors (or substrates) during sample processing was essential to ensure robust detection by LC-MS/MS. Cells were solubilised using LMNG (Lauryl Mannose Neopentyl Glycol)-containing buffer, a detergent shown previously to preserve labile ER-E3 complex interactions (*Schulz et al., 2017*). Immunoprecipitated E3-interactor complexes were washed and subsequently eluted from beads non-selectively by SDS to obtain the sample complexity necessary for subsequent comparative analyses (see below). All samples were prepared and processed for LC-MS/MS in parallel to facilitate comparative analysis (*Figure 1D*, see details in Materials and methods). Approximately 1600 individual protein groups were detected within each sample. In total, >2000 unique proteins were identified (*Supplementary file 1*, Table 2). To distinguish each E3's most relevant interactors, we adapted the Bait-Specific Control Group (BSCG) method described previously for multiple sample processing (*Keilhauer et al., 2015*; *Figure 1E*). This method defines the set of commonly detected interactors common as the 'background proteome' and normalises each sample to it to facilitate multi-sample comparison. By permitting relative fold-change and p-values to be determined for each identified protein, normalisation enables an enrichment to be assessed. To enrich for factors with relevant change, each putative E3 interactor was individually evaluated for: (1) p-value<0.05; (2) positive fold change; and (3) $\geq$2 unique peptides (*Figure 1E*). Commonly identified contaminants (*Supplementary file 1*, Table 3) were excluded and removed from the dataset. Proteins meeting all criteria were designated as 'high-confidence candidate interacting proteins' (HCIPs) (*Sowa et al., 2009*).

An inherent limitation of BSCG analysis is that candidate interactors detected in only one E3 sample evade classification as HCIPs because a relative fold change cannot be calculated. Since exclusive ER-E3 interactors represent high-value candidates, we also analysed raw data by calculating the semi-quantitative spectral index quantitation (SINQ) score for each sample (as described in *Trudgian et al., 2011*). Filtering for sample-exclusive, high SINQ score proteins in each sample identified an additional 28 interactors co-precipitating uniquely by specific ER-E3 (*Supplementary file 1*, Table 4). Merging the modified BSCG and SINQ analyses revealed over 431 interactions with 21 E3s (*Figure 1—figure supplement 1D*), the majority of which are unreported. This composite dataset represents a systematic attempt to define E3 complexes at the ER membrane.

## Interaction landscape of ER-resident E3s

From the 21 E3 ligases we comparatively examined, 218 different HCIPs that formed 403 interactions were identified (*Supplementary file 1*, Table 5). Hierarchical clustering of individual E3 interactomes (*Figure 2A*) revealed that HCIPs were both exclusive to and shared between ER-E3s. Visualising HCIP networks for E3s individually (*Figure 2—figure supplement 1*) revealed interactors brought down with varying degrees of confidence and linked to a diverse range of activities, as exemplified by the networks for both Hrd1 (*Figure 2B*) and RNF185 (*Figure 2C*). While ER-E3 raw abundance differed markedly, this did not correlate with the number of HCIPs identified (*Figure 2D*). Therefore, the number of interactions detected was a function of intrinsic E3 properties

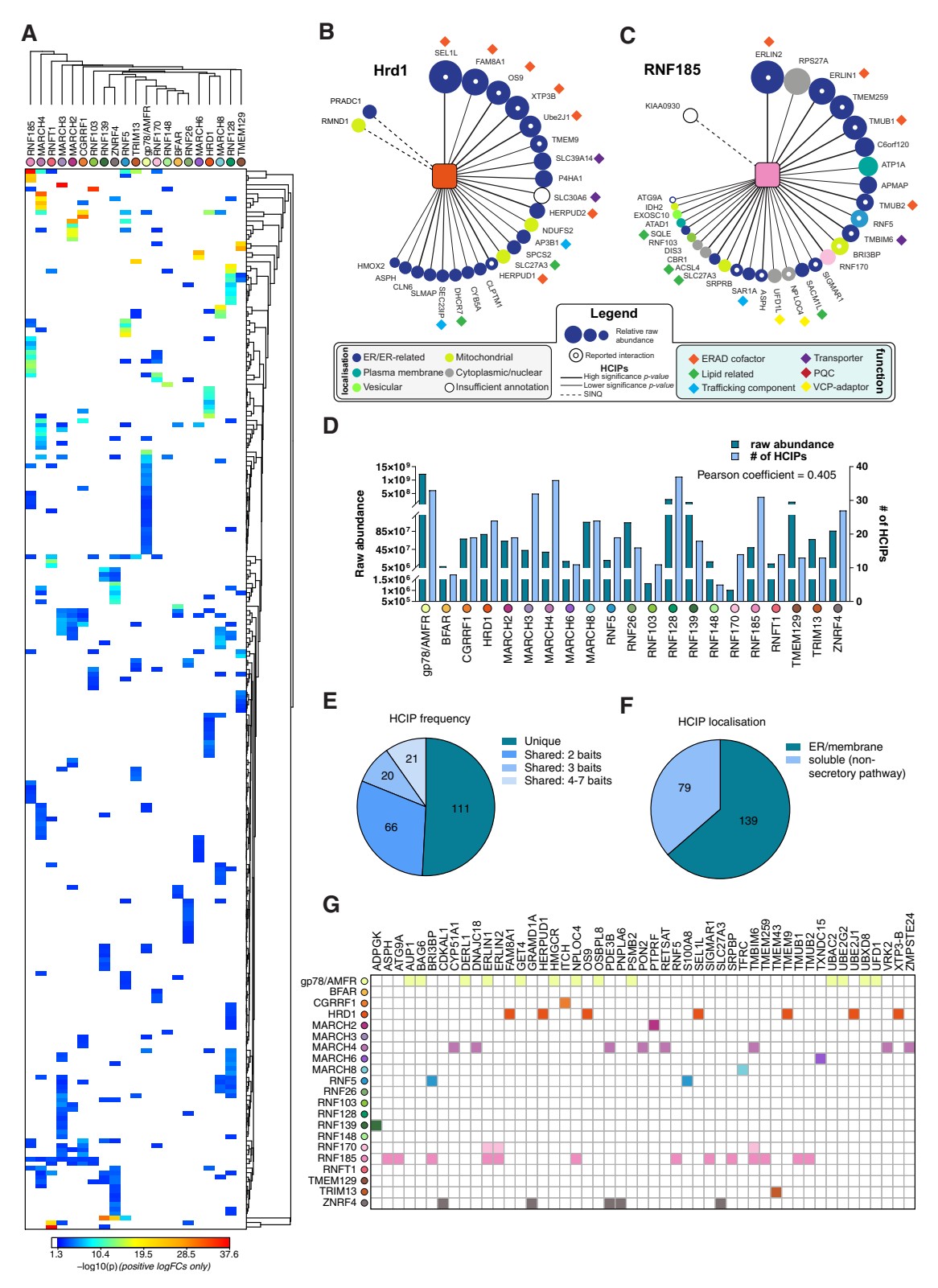

**Figure 2.** Interaction landscape of 21 ER-resident E3s. (**A**) Hierarchical clustering of the 21 E3s and their associated HCIPs represented as a heat map, where the colours of individual interactors correspond to their calculated p-values. Representative HCIP interaction wheels for (**B**) Hrd1 and (**C**) RNF185. Parameters represented are described in the adjoining legend. (**D**) Plot showing raw abundance (RA) and number of HCIPs determined for each E3 with the Pearson coefficient indicated. (**E**) Distribution of HCIP interactions with E3s as unique or shared. (**F**) Classification of HCIPs as ER/membrane or

*Figure 2 continued on next page*

*Figure 2 continued*

soluble, non-secretory pathway proteins as defined by presence of validated and predicted signal peptides, glycosylation sites, disulphide bonds, and transmembrane domains (UniProt). (G) E3-HCIP interactions identified previously in BioGRID 3.5 and BioPlex 3.0.

The online version of this article includes the following figure supplement(s) for figure 2:

**Figure supplement 1.** HCIP interaction network wheels for individual ER-E3s.

and not simply expression level. Importantly, over 50% of HCIPs (111/218) were significantly enriched by just one ER-E3 while those remaining associated with 2 (66/218), 3 (20/218) or >4 (21/218) different ligases (*Figure 2E*). HCIPs enriched by only one ER-E3 might represent specific cofactors or cognate substrates whereas interaction with multiple E3s could reflect proteins with generalised or adaptable functionality. HCIPs were not reported for RNF145, RNF150, TRIM59 and RNF13 because these baits were either not enriched or produced insufficient (≤2) peptide counts.

To concentrate subsequent analyses on interactions made at the ER, we searched the HCIP dataset for predicted protein features/domains associated with organelle targeting or residency, such as

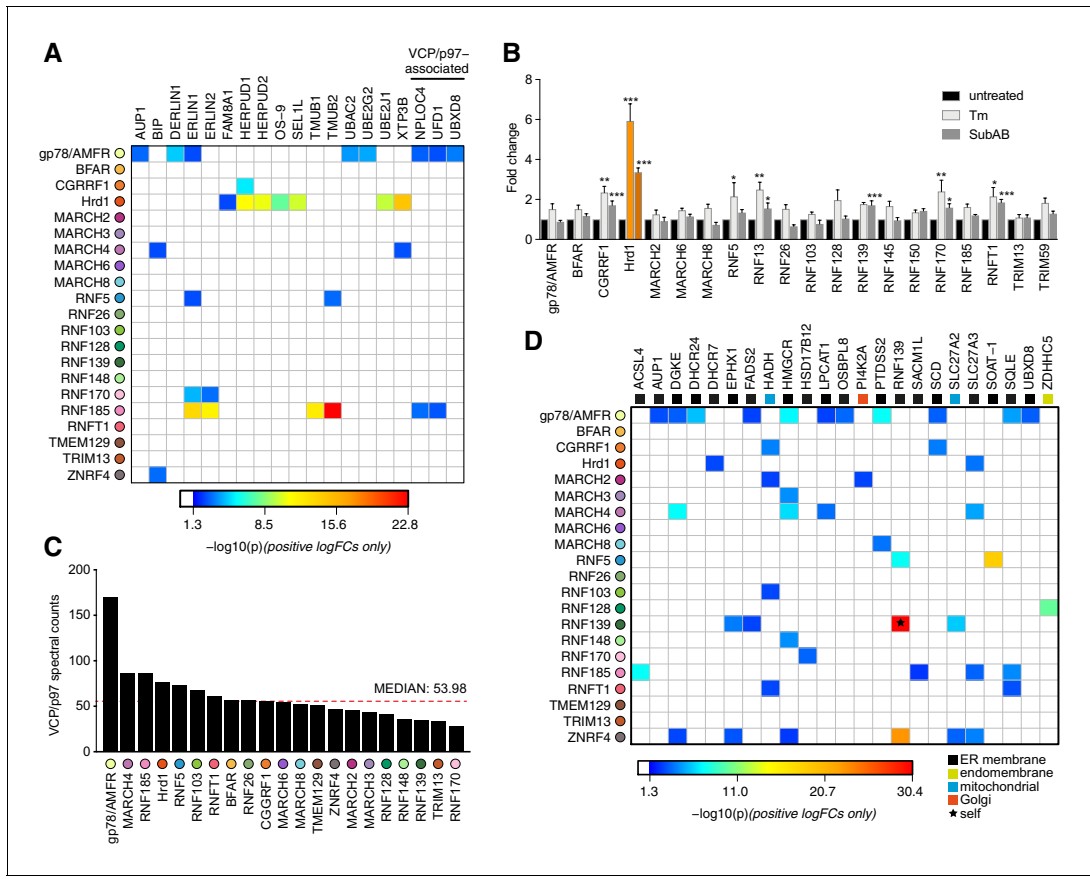

**Figure 3.** Functional associations of E3 and HCIPs. (A) Heat map depicting established ERAD components found as HCIPs with the panel of ER-resident E3s, with colours of individual interactors corresponding to their calculated p-values. (B) Transcriptional analysis of parental Flp-In293 cells determined by NanoString. Data depict fold change of E3 transcripts measured from tunicamycin-treated (Tm, 500 ng/mL, 8 hr) and SubAB-treated (SubAB, 10 ng/mL, 8 hr) cells when compared to untreated. Mean and S.E.M. are shown from three biological repeats (n = 3). *p<0.05, **p<0.01, ***p<0.001. Detailed statistical analysis can be found in *Supplementary file 1*, Table 12. Hrd1 is highlighted in orange for reference. (C) Absolute number of spectral counts detected for VCP/p97 for each ER-resident E3 determined by SINQ analysis. The dotted red line shows the median spectral counts for reference (D) Heat map representing the association between proteins involved in lipid regulation (synthesis; metabolism and transport) and E3 baits. Colours associated with individual interactors correspond to their calculated p-values.

The online version of this article includes the following source data and figure supplement(s) for figure 3:

**Source data 1.** Nanostring counts obtained for E3 transcripts in response to ER stressors used to generate *Figure 3B*.

**Figure supplement 1.** Validation of ER-resident E3 interactions.

TMDs, signal sequences, N-linked glycosylation or disulphide bonds (UniProt). Approximately two-thirds of HCIPs (139/218) contained features consistent with localisation to the endomembrane system, validating our enrichment strategy for ER-associated proteins (*Figure 2F*). The originality of the E3 interactions identified is reflected by their underrepresentation in current protein-protein interaction resources (BioGRID 3.5 and BioPlex 3.0), where only ~13% of E3-HCIP interactions were reported previously (*Oughtred et al., 2019*; *Huttlin et al., 2020*; *Figure 2G*). Over a third of these (20/57) were with either of the two extensively characterised ER-E3s, Hrd1 and gp78/AMFR. But importantly, large-scale interaction mapping efforts did identify and report nearly half (13/30) of the HCIPs from the less well-characterised RNF185 (*Huttlin et al., 2017*; *Huttlin et al., 2020*), which provides an additional level of assurance that these workflows can consistently identify bona fide ER-E3 interactors.

## ER-E3s, ERAD, and ER stress

E3 functionality at the ER is most commonly associated with the quality control (QC) process of ERAD (reviewed in *Claessen et al., 2012*; *Christianson and Ye, 2014*). We investigated whether any ER-E3s enriched for HCIPs implicated previously in ERAD, such as those that comprise the Hrd1 complex (e.g. SEL1L, UBE2J1, OS-9) (*Christianson et al., 2012*; *Hwang et al., 2017*). While all were associated with Hrd1 (*Figure 2B*), they were not HCIPs prominently enriched by other ER-E3s (*Figure 3A*). Established gp78/AMFR interactors including UBE2G2, Derlin1/DERL1, and UBAC2 (*Fang et al., 2001*; *Christianson et al., 2012*), also did not feature among HCIPs of other E3s (*Figure 2—figure supplement 1*, *Figure 3A*). Thus, factors previously linked with ERAD do not appear to have generalised functionality that is readily adopted by other ER-E3s, except for VCP/p97 (discussed below).

One way the unfolded protein response (UPR) resolves ER stress is by coordinated upregulation of Hrd1 (and its cofactors) to increase ERAD capacity (*Travers et al., 2000*). We investigated whether other ER-E3s respond similarly to ER stress by quantitatively monitoring their transcriptional changes in HEK293 cells treated with Tunicamycin (Tm) or the $AB_5$ family bacterial toxin Subtilase cytotoxin (SubAB) (*Paton et al., 2006*; *Supplementary file 1*, Table 6). CGRRF1, RNF13, RNF170 and RNFT1 transcript levels increased with acute ER stress (~2 fold) as reported previously (*Kaneko et al., 2016*), along with RNF5 (Tm only) and RNF139 (SubAB only) (*Figure 3B*). When compared to the ~6 fold change (for Tm) observed for Hrd1, however, any responsive contribution made by other E3s to ER stress resolution may be nominal. Of the 218 HCIPs identified, 24 are among the 278 previously collated targets of the UPR transcription factors XBP1, ATF6, and ATF4 (*Bergmann et al., 2018*), with a quarter (6/24) represented by the Hrd1 complex alone (*Supplementary file 1*, Table 7). Moreover, ER homeostatic maintenance did not require any individual ER-E3 since siRNA-mediated knockdowns of endogenous isoforms were not sufficient to induce the splicing of *XBP1* (*Figure 3—figure supplement 1A*), consistent with CRISPRi screens for ER stress induction (*Adamson et al., 2016*). Taken together, these findings are consistent with the unique positioning of Hrd1 among E3s to resolve proteotoxic ER stress (*Vitale et al., 2019*).

## Recruitment of VCP/p97 to ER E3 complexes

Binding and hydrolysis of ATP enables VCP/p97 to generate the force necessary to extract polypeptides from the ER membrane during ERAD (*Ye et al., 2003*; *Ye et al., 2004*). VCP/p97 enrichment could therefore reflect a need for substrate extraction and a role in ERAD, so we searched for reported ER-E3 and HCIP interactions with the AAA ATPase. VCP/p97 is recruited to protein complexes throughout the cell by factors that contain a UBX (Ub-regulatory X) domain or by linear sequences such as the SHP motif (also known as binding site 1, BS1), the VCP-interacting motif (VIM), or the VCP-binding motif (VBM) (*Meyer et al., 2012*). These domains/motifs can be found in ER-E3s (*Morreale et al., 2009*; *Ballar et al., 2006*), their cofactors (e.g. Derlins, VIMP) (*Greenblatt et al., 2011*; *Ye et al., 2005*) or ERAD-related enzymes (e.g RHBDL4) (*Fleig et al., 2012*). Surprisingly, only two ER-related HCIPs containing confirmed VCP-binding domains were found. FAF2/UBXD8 containing a UBX domain (*Mueller et al., 2008*) and Derlin1 containing a SHP domain (*Greenblatt et al., 2011*), were both associated with gp78/AMFR (*Figure 2—figure supplement 1*, *Figure 3A*) as previously reported (*Christianson et al., 2012*).

Differentiating bona fide recruitment of VCP/p97 from non-specific binding can be problematic in an IP-LC-MS/MS workflow because it is highly abundant, involved in a broad range of cellular processes, and can interact non-specifically. To address this, we compared VCP/p97 spectral counts determined for each E3 IP to assess whether any enriched the AAA ATPase. Among the highest were those for gp78/AMFR and Hrd1, in line with their established recruitment for ERAD, but RNF185, MARCH4 and RNF5 were also within the upper quartile (*Figure 3C*). The soluble VCP/p97 cofactors UFD1 and NPLOC4 were also HCIPs of RNF185 (*Figure 3A*, and discussed below), which lends additional support for this E3 in an ERAD-related role (*El Khouri et al., 2013*). RNF185 was recently highlighted among the set of prominent candidates identified in the VCP/p97 interactome (*Hülsmann et al., 2018*). Despite this, neither RNF185 nor its HCIPs contained canonical VCP/p97 binding domains/motifs that might justify recruitment. An unidentified factor or the presence of a non-canonical (or cryptic) VCP/p97 binding motif (*Buchberger et al., 2015*) could be responsible. Alternatively, VCP/p97 may be recruited indirectly to RNF185 through ubiquitin-binding cofactors (e.g. UFD1, NPLOC4) if the E3 is constitutively modified by ubiquitin chains (e.g. via auto-ubiquitination). However, the nature of this recruitment remains undetermined at present.

## Ubiquitin-related HCIPs and ER-E3s

ER-E3 HCIPs included factors implicated in Ub conjugation and binding, as well as those containing a Ub-like (UBL) domain. However, other than the two well-characterised E2-E3 pairs UBE2G2-gp78/AMFR and UBE2J1-Hrd1, our isolation protocol does not appear to have robustly preserved these interactions. As almost all E2s are soluble and bind with moderate or weak affinity to E3 RING domains during Ub transfer, their scarcity was not unexpected. Peptides from the ubiquitin precursor, ribosomal fusion protein RPS27A, were enriched by both RNF185 and gp78/AMFR, reflected by matched ubiquitin derived peptides quantified by LC-MS/MS, representing ubiquitin binding to or direct modification of the E3s. Hrd1 and RNF185 enriched UBL domain-containing proteins (and homologues) Herp/Herp2 (HERPUD1/HERPUD2) and TMUB1/TMUB2, respectively (*Figure 2B*; *Figure 2C*; *Figure 3A*). Both have been linked to ERAD; TMUB1 links Erlin1 to gp78 (*Jo et al., 2011b*) while Herp binds to FAM8A1 (*Schulz et al., 2017*) and activates Hrd1 through a cytoplasmic domain (*Kny et al., 2011*). Deubiquitinating enzymes (DUBs) were not among E3 HCIPs, which may not be surprising as the DUB interactome did not report interaction with any ER-E3s (*Sowa et al., 2009*). Some ER-E3s did co-precipitate (e.g. RNF185-RNF170, RNF185-RNF5, *Figure 2C*), and may reflect organisation consistent with coordinated or sequential ubiquitination as part of ERAD (*Morito et al., 2008*; *Zhang et al., 2015*) or alternatively, an E3-substrate relationship.

## ER-E3s and calcium-related HCIPs

Signalling from G-protein-coupled receptors (GPCRs) activates inositol 1,4,5-triphosphate (IP$_3$)-receptors (IP3Rs), causing this calcium-gated $Ca^{2+}$ channel to be ubiquitinated and turned over from the ER membrane (*Pearce et al., 2007*) by RNF170 and its cofactors ERLIN1/ERLIN2 (*Lu et al., 2011*; *Pearce et al., 2009*). We identified enrichment of the ERLIN1/2 heterodimer by RNF170 and RNF185, and ERLIN1 alone by RNF5 and gp78/AMFR (*Figures 2C* and *3A*, *Figure 2—figure supplement 1*). RNF170 is an HCIP of both RNF185 and gp78, which suggests ERLIN1/2 interactions could serve as a bridge for larger hetero-oligomeric E3 complexes. IP3R was only enriched by RNF170, in line with its previous identification as a cognate substrate and demonstrating that this methodology can also identify bona fide E3 substrates. Consistent with a larger hetero-oligomer, RNF170 and RNF185 share other HCIPs including the putative secreted factor c6orf120 and TMBIM6/BI-1/Bax-inhibitor 1 (*Figure 3—figure supplement 1B*). These form a $Ca^{2+}$ leak channel in the ER that protects cells from ER stress (*Bultynck et al., 2014*) by regulating $Ca^{2+}$ release and interacting with TMBIM3/GRINA (see *Rojas-Rivera et al., 2012*). RNF185 enriched for TMUB1/TMUB2 and TMEM259/Membralin (*Figure 2C*), a polytopic ER protein linked to motor neuron survival (*Yang et al., 2015*) that appears to have been erroneously assigned as part of the Hrd1-gp78 ERAD network (*Zhu et al., 2017*). RNF185 interactions were validated by co-expression and pulldown with S-tagged HCIPs (*Figure 3—figure supplement 1C*). Interestingly, the $Ca^{2+}$-load-activated $Ca^{2+}$ channel TMCO1, which prevents overfilling in the ER (*Wang et al., 2016*), was also enriched by RNF170. Collectively, RNF170 and RNF185 appear to associate with proteins linked to homeostatic maintenance of ER $Ca^{2+}$ levels related to ER stress and apoptosis.

## E3s and lipid-related HCIPs

Sterols and fatty acids are produced through coordinated biosynthetic reactions at the ER membrane. 3-hydroxy-3-methylglutaryl-CoA-reductase (HMGCR) and squalene epoxidase (SQLE), are among the best examples of rate-limiting enzymes degraded by ER-E3s through negative feedback to regulate biosynthetic activity. 23 different HCIPs were involved in the biosynthesis and regulation of cholesterol, fatty acids, or phospholipids, with 80% annotated as TMD-containing proteins residing in the ER (*Figure 3D*). Sterol and fatty acid-related HCIPs were enriched by 16 different E3s with nearly a quarter of all interactions (11/48) made with gp78/AMFR. Among the HCIPs of gp78/AMFR was HMGCR, previously reported as its substrate (*Jo et al., 2011a*). Conventionally, lipid-related HCIPs associating with E3s would do so as substrates for regulatory ERAD processes and may include DHCR7 (Hrd1), ACSL4 (RNF185) or SOAT1 (RNF5). Additionally, ZDHHC5 (RNF128) is a palmitoyl-acyltransferase which itself is palmitoylated (*Kang et al., 2008*; *Martin and Cravatt, 2009*) and involved in endosome-Golgi trafficking (*Breusegem and Seaman, 2014*). Notably, it is one of the few palmitoyl transferases localised to the endosomal system, consistent with RNF128 (GRAIL) also reportedly present beyond the ER in endocytic compartments (*Anandasabapathy et al., 2003*).

## RNF26 is unstable and degraded by the ubiquitin-proteasome system (UPS)

ER-E3s co-precipitated uncharacterised proteins with high specificity and abundance, which may reflect essential functionality at the ER. One example is the HCIPs and complexes formed by RNF26, an ER-E3 implicated previously in innate immune signalling (*Qin et al., 2014*) and organisation of perinuclear endosomes (*Jongsma et al., 2016*). RNF26 is integrated into the ER membrane through six predicted TMDs (*Figure 4A*). Its canonical C3-H-C4-type RING domain lying near its C-terminus is evolutionarily conserved (*Figure 4—figure supplement 1A*) and shares sequence and positional similarity with the nuclear SUMO-targeted Ub ligase (StUbL) RNF4 (*Liew et al., 2010*; *Plechanovová et al., 2012*), Inhibitor of Apoptosis Proteins (IAPs, [*Gyrd-Hansen et al., 2008*]), and MDM2 (*Linke et al., 2008*; *Figure 4B*), all of which form homo-/heterodimers through their RING domains. Despite efforts and consistent with published data (*Jongsma et al., 2016*), endogenous RNF26 protein could not be detected from whole cell lysates, even though relative mRNA levels in HEK293 cells were comparable to other E3s such as Hrd1 (*Figure 4—figure supplement 1B*). Consequently, we used DOX-induced expression of FH-RNF26$_{WT}$ to facilitate detection by immunoblotting, which was enhanced by treatment with MG132 (*Figure 4C*, top panel, *Figure 1—figure supplement 1A*). Stabilisation by MG132 reflected intrinsic instability of RNF26 and suggested constitutive disposal by ERAD or an ERAD-like process. RNF4 conjugates Ub using a penultimate tyrosine (Y189) to engage the E2-Ub thioester (*Plechanovová et al., 2011*). This aromatic residue is conserved in RNF26 (Y432, *Figure 4A and B*, *Figure 4—figure supplement 1A*), where a neutralising mutation (Y432A) stabilises FH-RNF26 protein levels and renders it insensitive to MG132 (*Figure 4C*, middle panel). Radiolabel pulse-chase assays confirmed the increased stability of newly synthesised FH-RNF26$_{Y432A}$ compared to FH-RNF26$_{WT}$ ($t_{1/2}$ ~ 2 hrs. vs. <0.5 hr, *Figure 4D*). Expression of FH-RNF26$_{Y432A}$ also enabled endogenous RNF26 to be co-precipitated and detected (*Figure 4E*), consistent with the formation of stable and inactive heterodimers.

Like other E3 dimers, the increased stability afforded by MG132 and the E2-Ub activating mutant (Y432A) is consistent with RNF26 auto-ubiquitination. Accordingly, immunoprecipitated FH-RNF26$_{WT}$ produced FLAG-immunoreactive ladders and high-molecular weight smears, enhanced by MG132 and collapsed by the non-selective DUB Usp21 (*Figure 4F*). In vitro ubiquitination assays using immunopurified FH-RNF26 (WT or Y432A) and recombinant UbcH5a, faithfully recapitulated these observations (*Figure 4—figure supplement 1C*). Curiously however, immunoprecipitated FH-RNF26$_{WT}$ subjected to a panel of linkage specific DUBs indicated RNF26 modification by Ub chains not conventionally linked with degradation, including K33-, K63- and K29-linkages as well as multiple mono-ubiquitination (*Figure 4—figure supplement 1D*). Linkage diversity in RNF26 ubiquitination may indicate effects not only on turnover, but also on scaffolding and interaction with other proteins.

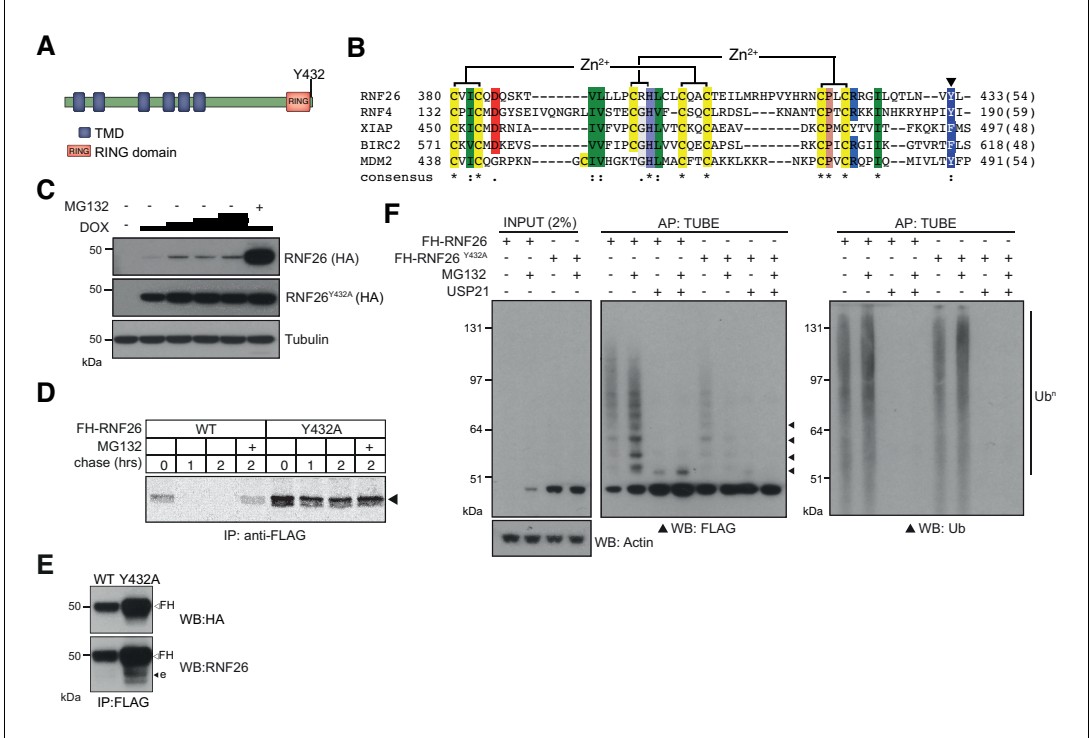

**Figure 4.** Characterisation of the RNF26 ubiquitin ligase complex. (A) Domain organisation of the human RNF26 protein. (B) Protein sequence alignment of the RING domain and C-terminus of RNF26 with those of human RNF4 (P78317), XIAP (P98170), BIRC2 (Q13490) and MDM2 (Q00987). Conserved residues are demarcated according to the Rasmol colour scheme. (C) DOX titration of Flp-In293 cells stably expressing FH-RNF26$_{WT}$ or FH-RNF26$_{Y432A}$ ± MG132 with lysates separated by SDS-PAGE and resulting western blots probed for RNF26 (anti-HA) and tubulin. (D) $^{35}$S-Met/Cys pulse-chase assay (0, 1, 2 hr) of DOX-induced Flp-In293 cells stably expressing FH-RNF26$_{WT}$ or FH-RNF26$_{Y432A}$,±MG132 and immunoprecipitated by anti-FLAG agarose. (E) Co-immunoprecipitation of endogenous RNF26 from DOX-induced Flp-In293 cells stably expressing FH-RNF26$_{WT}$ or FH-RNF26$_{Y432A}$ by anti-FLAG beads. Detection of FLAG-HA (FH) and endogenous (e) RNF26 by western blot using the indicated antibodies. (F) TUBE pulldowns from FH-RNF26$_{WT}$ or FH-RNF26$_{Y432A}$ Flp-In293 cell lysates,±MG132 and Usp21.

The online version of this article includes the following source data and figure supplement(s) for figure 4:

**Figure supplement 1—Source data 1.** Nanostring data of E3 transcript abundance relative to CNX used to generate *Figure 4—figure supplement 1B*.
**Figure supplement 1.** RNF26 ubiquitination.

## Discovery and identification of RNF26 interactors

The intrinsic instability of RNF26$_{WT}$ may have led low abundance interactors to be below detection thresholds, and so IP-LC-MS/MS was also performed using FH-RNF26$_{Y432A}$, which was then introduced into the BSCG and SINQ analyses (*Figure 5A*). The inclusion of FH-RNF26$_{Y432A}$ increased the total number of high-confidence protein-protein interactions from 431 to 460. A comparison of RNF26$_{WT}$ and RNF26$_{Y432A}$ HCIPs revealed many of the same interactors, albeit now with more specific enrichment (i.e. higher significance p-values) (*Figure 5B*, *Supplementary file 1*, Table 8). We prioritized TMD-containing and ER-related HCIPs for validation, selecting first among those enriched by both forms. Prominent HCIPs of both RNF26 forms included TMEM43/LUMA (*Figure 5B*, *Figure 5—figure supplement 1A*), an evolutionarily ancient multi-spanning membrane protein present in both the ER and INM (*Bengtsson and Otto, 2008*; *Dreger et al., 2001*), and ENDOD1 (endonuclease domain containing-1), an uncharacterised polytopic protein containing a putative DNA/RNA non-specific nuclease domain (*Figure 5B*, *Figure 5—figure supplement 1A*). RNF26$_{Y432A}$ also enriched for TMED1, an atypical member of the p24 cargo receptor family (*Jenne et al., 2002*), an ER membrane-shaping factor TMEM33 (*Urade et al., 2014*), as well as the ERAD and lipid droplet-related proteins UBXD8 and AUP1 (*Figure 5A and B*, *Figure 5—figure supplement 1A*).

To validate interactions and gain insight into quaternary structures, S-tagged HCIPs were co-expressed with FH-RNF26$_{WT}$ and the resulting co-precipitation profiles compared. FH-RNF26 was reproducibly brought down by S-tagged TMEM43, TMED1, AUP1 and UBXD8, with the latter

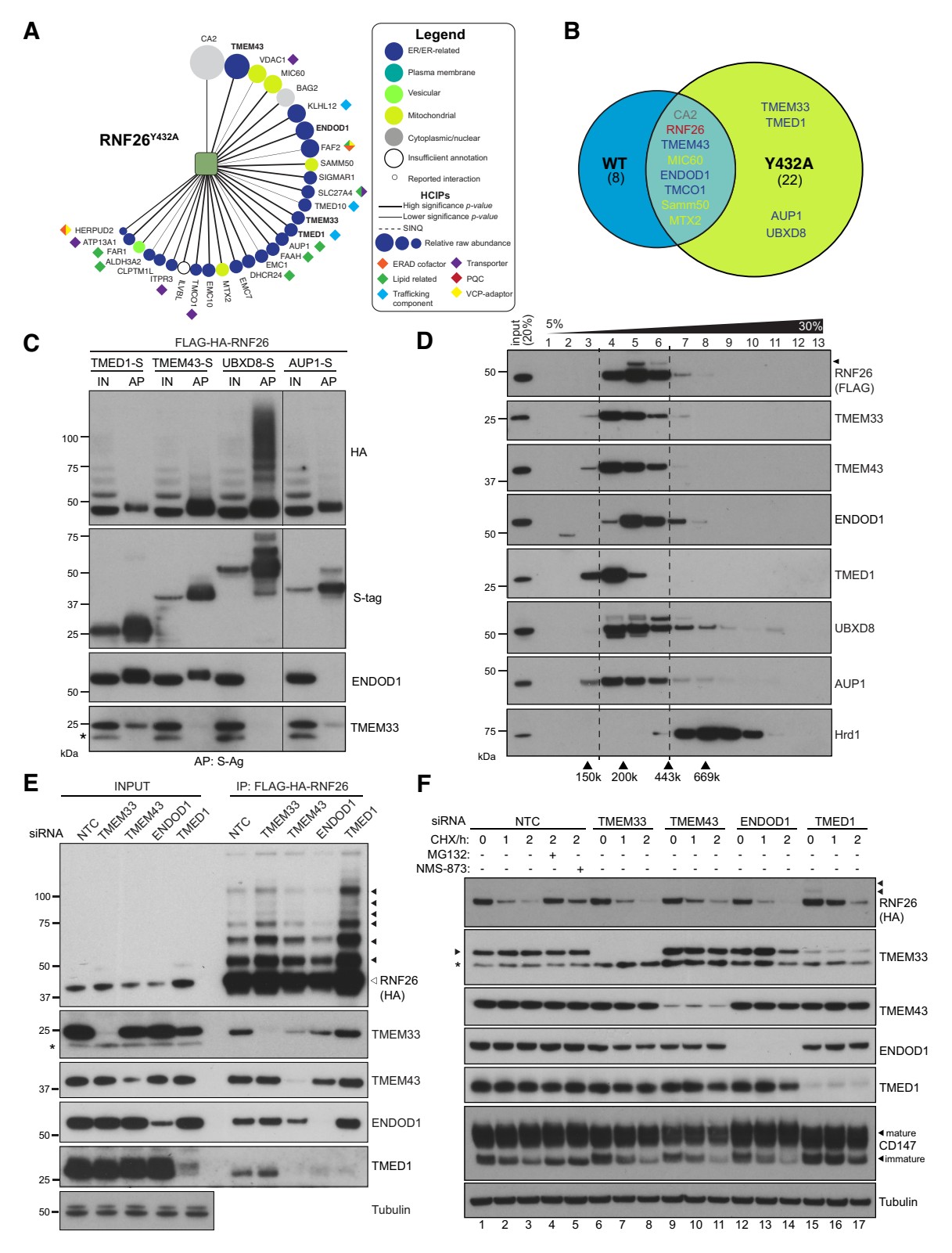

**Figure 5.** RNF26 assembles with HCIPs in the ER. (**A**) HCIP interaction network wheel for FH-RNF26_{Y432A}. Legend described in *Figure 2—figure supplement 1*. (**B**) Venn diagram of HCIPs identified by LC-MS/MS for FH-RNF26_{WT} and FH-RNF26_{Y432A}. ER-resident HCIPs are indicated in blue. (**C**) Co-precipitation of FH-RNF26_{WT} from stable Flp-In293 cells by transiently expressed S-tagged HCIPs (TMED1, TMEM43, UBXD8 and AUP1). Cells were solubilised in 1% LMNG and protein complexes affinity purified from the resulting lysates by S-protein agarose. Western blots of affinity purified

*Figure 5 continued on next page*

Figure 5 continued

material (AP) and input lysate (IN, 20%) were probed with antibodies recognising RNF26 (HA), HCIPs (S-tag), TMEM33 and ENDOD1. (D) Velocity sedimentation of FH-RNF26 complexes from 1% LMNG-solubilised lysates on a sucrose gradient (5–30%), with individual TCA-precipitated fractions (1-13) subsequently separated by SDS-PAGE and the resulting western blots probed for the indicated proteins. Ubiquitinated forms of RNF26 are indicated by black arrowheads. Molecular weight of gel filtration standards solubilised and sedimented in equivalent buffer conditions are shown for comparison beneath the Hrd1 blot, which in turn serves to highlight a complex of a different mass. (E) siRNA-mediated knockdown of HCIPs in FH-RNF26$_{WT}$ Flp-In293 cells alters interaction profiles. RNF26 complexes immunoprecipitated by anti-FLAG agarose were separated by SDS-PAGE with resulting western blots probed by antibodies for RNF26 and the indicated HCIPs. Tubulin was used as a loading control. Ubiquitinated forms of RNF26 are indicated by black arrowheads. (F) Cycloheximide (CHX) chase assays (100 μg/ml; 0, 1, 2 hr) of FH-RNF26$_{WT}$ Flp-In293 cells (DOX, 1 μg/mL, 18 hr) knocked down for individual HCIPs by siRNAs with the resulting western blots probed for the indicated antibodies. MG132 (10 μM, 2 hr) and NMS-873 (10 μM, 2 hr) were included with NTC samples where indicated. Ubiquitinated forms of RNF26 are denoted by black arrowheads. Mature and immature forms of CD147 are indicated by black arrowheads.

The online version of this article includes the following figure supplement(s) for figure 5:

**Figure supplement 1.** Validation of RNF26 interactions with HCIPs.

exhibiting a high molecular weight smear of RNF26 reminiscent of polyubiquitinated forms (*Figure 5C*). For reasons that are not clear, exogenous expression of S-tagged ENDOD1 and TMEM33 proved difficult to detect, however S-tagged TMEM43 and TMED1 were able to bring down endogenous ENDOD1 and TMEM33 (TMED1 only), indicating that these HCIPs comprised larger macromolecular complexes. Neither ENDOD1 nor TMEM33 were co-precipitated by S-tagged AUP1 or UBXD8, suggesting the formation of heterogeneous complexes by RNF26. Velocity sedimentation of lysates from FH-RNF26$_{WT}$ expressing cells revealed a profile of a complex migrating between ~200–300 kDa (fractions 4–6) with endogenous TMEM43, TMED1, ENDOD1, TMEM33, UBXD8 and AUP1 co-sedimenting with FH-RNF26, albeit with varying degrees of overlap within fractions (*Figure 5D*). Stoichiometry within the complex is not yet fully appreciated and must be considered carefully as the induced expression of FH-RNF26 could impact its native arrangement. The robustness of RNF26-HCIP interactions was assessed by performing IPs in Triton X-100 (TX-100) as well as the milder LMNG; detergents which differ in their ability to stabilise ER membrane protein complexes (*Schulz et al., 2017*; *Figure 5—figure supplement 1B and C*). TMEM43 co-precipitated by FH-RNF26$_{Y432A}$ was independent of detergent conditions while other HCIP interactions were compromised to varying degrees in TX-100. These data support the formation of one or more heteromeric complexes containing RNF26 and HCIPs with a key interaction likely made through TMEM43.

To investigate organisation of RNF26 complexes in greater detail, validated siRNAs targeting HCIPs (*Figure 5—figure supplement 1D–G*) were introduced into FH-RNF26$_{WT}$ expressing cells and co-precipitation profiles monitored. Silencing TMEM43 disrupted the ability of FH-RNF26$_{WT}$ to co-precipitate other HCIPs, without substantially altering total cellular levels (*Figure 5E*). A reduction in immunoprecipitated RNF26 (both unmodified and ubiquitinated) resulted from ENDOD1 knockdown, which might underlie the lower levels of other HCIPs in RNF26 pulldowns, in particular TMED1. Although the pattern of HCIPs co-precipitated by RNF26 was unaffected by depletion of either TMEM33 or TMED1, detection of both unmodified and ubiquitinated forms of RNF26 was markedly enhanced with the loss of TMED1. These profile changes, together with the fact that the RNF26-TMEM43 interaction is resistant to solubilisation in TX100, indicate TMEM43 plays a key role in RNF26 complex formation, while ENDOD1 and TMED1 exert influence over RNF26 abundance.

We next asked whether HCIPs influenced RNF26 stability. In cycloheximide (CHX) chase assays, siRNA-mediated depletion of TMEM43, TMEM33 or ENDOD1 did not markedly stabilise RNF26 nor did they alter turnover of immature CD147 (*Figure 5F*), a well-characterised Hrd1-dependent ERAD substrate (*Tyler et al., 2012*). TMED1 knockdown resulted in a modest slowing of RNF26 degradation leading to an increase in overall RNF26 levels (*Figure 5E*). A similar attenuation of CD147 degradation also suggests TMED1 could somehow be indirectly influencing ERAD mediated by Hrd1 (see Discussion). In contrast to RNF26, its HCIPs appeared stable and were relatively unaffected by loss of their counterparts (*Figure 5F*), even though in some cases they were no longer in a complex (*Figure 5E*). Of note, VCP/p97 inhibition by NMS-873 only partially restored RNF26 while stabilising CD147 equivalently to MG132 (*Figures 5F* and *4A*). RNF26 was not among those E3s that enriched VCP/p97 (*Figure 3C*), potentially distinguishing its mechanism of turnover from that of a canonical,

misfolded ERAD substrate. Neither MG132 nor NMS-873 increased basal levels of RNF26 HCIPs, consistent with relative stability in the ER membrane. These data indicate RNF26 instability is intrinsic and unaffected by its interactions with more stable HCIPs.

## RNF26 interactors regulate STING-dependent innate immune signalling

The immune signalling adaptor STING (STimulator of INterferon Genes, also known as MITA/ TMEM173) is a polytopic ER-resident protein activated by the cyclic dinucleotide cGAMP, a product of cyclic GMP-AMP-synthase (cGAS) generated in response to the presence of cytosolic double-stranded DNA (*Bridgeman et al., 2015*; *Figure 6A*). Activation of dimerised STING by cGAMP triggers higher order oligomerisation in the ER (*Shang et al., 2019*), type I interferon (IFN) signalling through recruitment of TBK1 and IRF3, and subsequent efflux into ERGIC-derived vesicles (*Ishikawa et al., 2009*; *Dobbs et al., 2015*) leading to its eventual turnover by p62/SQSTM-dependent autophagy (*Prabakaran et al., 2018*). In the ER, STING is a target for ubiquitination by multiple E3s including RNF26 (*Qin et al., 2014*; *Zhong et al., 2009*; *Wang et al., 2014*), which are reported to modulate its IFN signalling capability through UPS-dependent degradation. STING and RNF26 reportedly interact through their TMDs and the absence of RNF26 (or its RING domain) unexpectedly increases STING turnover and attenuates IFN signalling (*Qin et al., 2014*). We therefore considered whether the HCIPs of RNF26 might also modulate STING itself or signalling through it. We

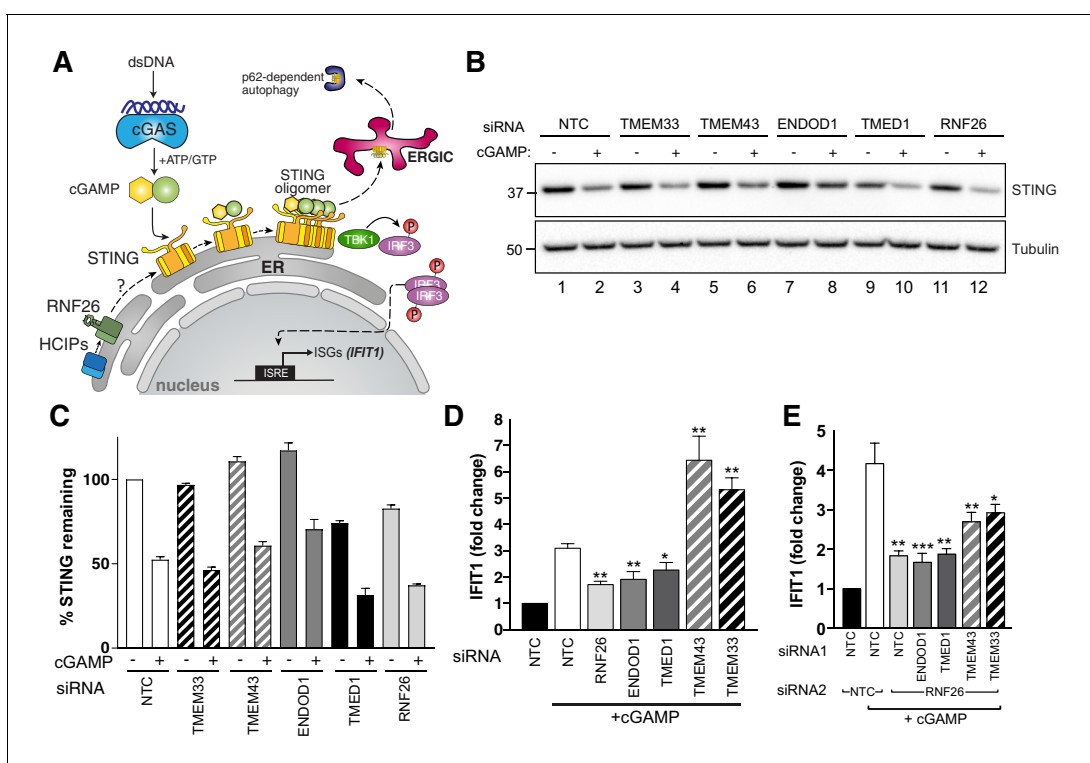

**Figure 6.** RNF26 and its interactors modulate STING-dependent innate immune signalling. (**A**) Diagram of the cGAS-STING signalling pathway. (**B**) Representative western blot of cGAMP-treated Flp-In293 cells transfected with siRNA targeting RNF26 and HCIPs (TMEM33, TMEM43, ENDOD1, TMED1) and probed for STING and tubulin. (**C**) Quantification of 3 biological replicates for (**b**) with mean and S.E.M. shown (n = 3). (**D**) qRT-PCR for *IFIT1* and *GAPDH* from cGAMP-treated Flp-In293 cells (5 µg/ml, 6 hr) transfected with siRNAs targeting RNF26, ENDOD1, TMED1, TMEM43 and TMEM33, along with a non-targeted control (NTC). Normalised *IFIT1* levels in cGAMP-treated cells are shown relative to their untreated counterpart for each siRNA. Mean and S.E.M are shown for at least four biological replicates. (**E**) Same as (**D**) but including siRNA targeting RNF26 along with HCIPs. Mean and S.E.M are shown for four biological replicates. For all statistical analysis, *p<0.05, **p<0.01, ***p<0.001. Details of statistical analysis are in *Supplementary file 1*, Table 12.

The online version of this article includes the following source data and figure supplement(s) for figure 6:

**Source data 1.** STING protein quantification and *IFIT1* fold change data used to generate *Figure 6*.

**Figure supplement 1.** Modulation of the interferon response by RNF26 and its HCIPs.

**Figure supplement 1—source data 1.** *IFIT1* fold change and RNF26 transcript abundance data used to generate *Figure 6—figure supplement 1*.

observed that like RNF26 knockdown, TMED1 siRNAs reduced STING protein levels by ~30% whereas those targeting TMEM43, ENDOD1 or TMEM33 had little to no impact (*Figure 6B and C*). Directly introducing cGAMP activates STING and promotes its trafficking and subsequent degradation, reflected by reduced detection of STING protein (*Figure 6B and C*). Reductions in STING abundance were proportional in all knockdowns (~50%), indicating that the influence of RNF26 and TMED1 did not depend on whether or not STING could be activated.

To ascertain whether RNF26 HCIPs also influence STING-mediated signalling in response to cGAMP, we monitored transcription of the interferon-stimulated gene (ISG) *IFIT1* (interferon-induced protein with tetratricopeptide repeats 1) after silencing HCIPs alone and together with RNF26. STING activation by cGAMP increased *IFIT1* transcription ~3–4 fold in WT cells (*Figure 6D and E*) and did so in a dose-dependent manner (*Figure 6—figure supplement 1A*). RNF26 knockdown (*Figure 6D*, *Figure 6—figure supplement 1B*) attenuated the increase in *IFIT1*, consistent with previous findings (*Qin et al., 2014*). Expression of FH-RNF26$_{Y432A}$ also inhibited *IFIT1* upregulation (*Figure 6—figure supplement 1C*), demonstrating that the *IFIT1* response requires ubiquitination by RNF26 and that the Y432A mutant functions as a dominant negative. Knockdown of either TMED1 or ENDOD1 also dampened the cGAMP-induced *IFIT1* response (*Figure 6D*), but lower STING levels only coincided with TMED1 loss and not ENDOD1 (*Figure 6B and C*), suggesting these two HCIPs impact STING and signalling through it, by different mechanisms. In marked contrast, depleting cells of either TMEM43 or TMEM33 enhanced *IFIT1* responses to cGAMP, resulting in 5–7-fold increases that were nearly twice the magnitude elicited from CTRL cells (*Figure 6D*). These results were recapitulated with an independent set of siRNAs (*Figure 6—figure supplement 1D*). Co-depleting RNF26 along with TMED1 or ENDOD1 did not attenuate *IFIT1* responses further, and co-depleting with either TMEM43 or TMEM33 reduced the enhanced *IFIT1* responses by ~50% (*Figure 6E*), indicating that the impact of these HCIPs on STING activation was at least in part through RNF26. These data describe four novel modulators of STING-dependent immune signalling and define the RNF26 complex as an immunoregulatory unit.

## Discussion

The ER accommodates a range of functional modalities; many of which fall under the regulatory remit of ubiquitination by resident E3-containing complexes. Comparative proteomic strategies have generated extensive and informative interaction networks of ubiquitination machinery (*Sowa et al., 2009*; *Bennett et al., 2010*). We have applied those principals to expand the functional ERAD modules described for Hrd1 and gp78/AMFR (*Christianson et al., 2012*) by mapping over 450 interactions to form the landscapes for 21 ER-E3s. The ER membrane is increasingly appreciated as a diverse site of regulation for metabolic and homeostatic processes, which is reflected in the diversity of interactions made by resident E3s.

### HCIPs represent cofactors and substrates of ER-E3s

ER-E3s assemble multi-subunit complexes from protein-protein interactions made through lipid bilayer, lumenal, and cytosolic contacts, rendering these units highly adaptable at conjugating ubiquitin proximal to the membrane. Because ER-E3s appear structurally and topologically diverse, the minimal redundancy their HCIP networks exhibit (*Figure 2A*) is consistent with assembling complexes from dedicated cofactors that would favour selective cellular responsibilities. Nearly 70% of HCIPs identified contain predicted TMDs or signatures associated with ER residency, among which will be both E3 cofactors and substrates. E3 cofactors are exemplified by Hrd1 interactors such as SEL1L and FAM8A1, but also by those validated for RNF26 (*Figure 5*) and proposed for RNF185 (*Figure 3—figure supplement 1B*). Substrates are typified by the Wnt receptor Evi/WLS/GPR177, which after being identified early in this study was established as novel target for regulated ERAD by CGRRF1 (*Glaeser et al., 2018*). Other promising candidate substrates might include DHCR7, the terminal enzyme of cholesterol biosynthesis found with Hrd1 and a validated ERAD substrate (*Prabhu et al., 2016*; *Huang et al., 2018*) and ACSL4, an RNF185 HCIP catalysing long chain polyunsaturated CoA synthesis (*Küch et al., 2014*) which global turnover studies indicate has a short half-life (*McShane et al., 2016*). SLC39A14/ZIP14, a stress-regulated $Zn^{2+}$ and $Mn^{2+}$ transporter at the plasma membrane, was identified as a Hrd1 HCIP. As it was detected under conditions without ER stress, Hrd1 may be responsible for constitutive degradation of ZIP14 to maintain metal

homeostasis. Like these, additional E3-substrate relationships are likely to be present within the dataset but will require detailed kinetic analyses to confirm them. Identifying and differentiating cognate cofactors from substrates of E3s remains a significant challenge in ubiquitin biology. Proteomics-based studies such as this, cannot readily distinguish E3 substrates from cofactors participating in substrate recruitment, complex assembly, localisation or ubiquitin conjugating activity. As the example of RNF26 HCIPs illustrates, biochemical, genetic and functional validation remains essential to accurately define complexes within each individual network. Although not providing comprehensive insight into all ER-E3 complexes that form, our analysis establishes a starting point from which to systematically elucidate their molecular organisation and functional responsibilities.

## Ubiquitin linkages and machinery associated with ER-E3s

Ubiquitination at the ER is often envisaged in terms of conventional ERAD, where associated ubiquitin linkages (e.g. K48, K11) added to substrates by one or more ER-E3s facilitates recognition by ubiquitin-binding proteins associated with VCP/p97 (*Locke et al., 2014*) and proteasomes (*Xu et al., 2009*). But our findings highlight the potential of some resident E3s to function differently. For example, RNF26 reportedly modifies STING with K11 chains (*Qin et al., 2014*) but we showed its own turnover from the ER is rapid and is associated with K33-, K63- and/or K29-linked ubiquitin chains not conventionally associated with degradation or targeting to proteasomes. K29-linkages can be part of heterotypic branched and mixed Ub chains (*Kristariyanto et al., 2015*), which hallmark some ERAD substrates (*Leto et al., 2019*). What role linkage heterogeneity plays in RNF26 turnover is not yet clear but might indicate linkages other than K48- and K11- also possess the ability to target proteins from the ER to proteasomes (*Xu et al., 2009*).

E2s are determinants of poly-ubiquitin linkages while associated DUBs trim and prune them with linkage-specificity, yet most pairwise relationships with E3s remain undefined. As there are ~40 E2s and nearly ~100 DUBs in the human proteome, we would expect one (or more) to partner with each E3 to expand its substrate range. However, we instead found that E2s and DUBs were not prominent among HCIPs. While key interactions for ERAD were confirmed (UBE2J1-Hrd1 [*Christianson et al., 2012*; *Mueller et al., 2008*] and UBE2G2-gp78 [*Das et al., 2009*]), the often transient, low affinity E3-E2 and E3-DUB interactions appear not to have been widely preserved. Yeast-2-hybrid screens used to define human E2-E3 pairs report ER-E3s (e.g. RNF26, BFAR, RNF5) as interactors of multiple E2s from prey libraries (*van Wijk et al., 2009*; *Markson et al., 2009*), but these are yet to be validated in vivo and were not HCIPs. Similarly, functional roles in ERAD were identified for USP13, Atx3, VCPIP and USP19, but ER-E3s were not identified among the DUB interactome (*Sowa et al., 2009*). Detailed mapping of the ubiquitin linkage landscape attributable to individual ER-E3s and the UPS machinery required, remains an outstanding question for future studies.

## The specialised role of Hrd1 among ER-E3s

ERAD is responsible for misfolded protein disposal during ER stress to help restore organelle homeostasis. Induction of any ER-E3 by pro-survival UPR branches might have signalled complementary contributions to ERAD and ER stress resolution provided by the Hrd1 complex. Although ER stressors upregulated some ER-E3s (e.g. CGRRF1, RNF170, RNF5; *Figure 3B*, *Kaneko et al., 2016*), their upregulation was modest relative to that observed for Hrd1. This is consistent with the observation that Hrd1 was the only E3 upregulated by direct activation of either the XBP1 or ATF6 transcription factors (*Shoulders et al., 2013*). Moreover, there were only 24 HCIPs among known UPR target genes (*Bergmann et al., 2018*) with at least a quarter belonging to the Hrd1 complex (*Supplementary file 1*, Table 7, *Christianson et al., 2012*). Thus, evolutionary expansion of E3s in the mammalian ER has not extensively supplemented adaptive stress-resolving capabilities functionally redundant with Hrd1, consistent with ubiquitination by the Hrd1 complex being essential for survival under proteotoxic ER stress conditions (*Yagishita et al., 2005*; *Vitale et al., 2019*). With misfolding of secreted rather than membrane proteins being the principal instigators of ER stress, an exclusivity of access to lumenal substrates via SEL1L offers an explanation why Hrd1 appears indispensable. Instead, other ER-E3s already known to oversee specific metabolic or regulatory functions consistent with higher order organismal functions, appear more likely to have evolved more selective client ranges of integral membrane proteins. ER-E3s may only hold strategic importance in selected cells or tissues, where a particular substrate/s are physiologically relevant. The enrichment of VCP/

p97 only by only some ER-E3s (e.g. Hrd1, gp78, RNF185) supports a model where ERAD-like degradation is only one of the processes E3s oversee at this membrane interface.

## RNF26 is an intrinsically unstable ER-E3

We found RNF26 to be turned over rapidly from cells by the proteasome (*Figure 4C and D*). This coincided with its auto-ubiquitination, which was abolished in the Y432A mutant (*Figure 4F*), consistent with the function of this penultimate tyrosine residue in RNF4 - an E3 whose C-terminus shares a high degree of sequence similarity with that of RNF26. Tyr189 of RNF4 enables E2-ubiquitin oxyester hydrolysis by binding the Ile44 hydrophobic patch of ubiquitin at the RNF4 dimeric interface (*Plechanovová et al., 2012*). RNF4 functions as a dimeric E3 and likewise, we were able to detect endogenous RNF26 following immunoprecipitation of FH-RNF26$_{Y432A}$ (*Figure 4E*). It is important to note that detection of endogenous RNF26 has reportedly been problematic (*Jongsma et al., 2016*), something we also encountered. This is unlikely to be attributable to the cell lines used, as RNF26 transcript is more prevalent in HEK293 cells than the readily detectable Hrd1 (*Figure 4—figure supplement 1B*). Its short half-life together with a lack of affinity reagents with sufficient sensitivity are likely reasons currently limiting more robust detection of RNF26. Future development of tools able to isolate and/or detect native RNF26 would enable our observations on stability and complex formation to be confirmed at the endogenous level.

## cGAS-STING pathway regulation by RNF26

Chronic activation of STING is linked to autoimmune and auto-inflammatory disorders (*Cai et al., 2014*), denoting an imperative for tight regulation and fine control over this signalling cascade. Ubiquitination offers spatiotemporal and post-translational control of STING abundance, activation and consequently, signalling. Multiple E3s, including RNF26, reportedly ubiquitinate STING while in the ER, and after it is trafficked into endolysosomal vesicles following cGAMP-induced oligomerisation (reviewed in *Hopfner and Hornung, 2020*). RNF26 knockdown lowered STING levels and dampened *IFIT1* upregulation (*Figure 6B–D*). Expression of FH-RNF26$_{Y432A}$ exerted the same effect on *IFIT1* levels (*Figure 6—figure supplement 1C*), placing RNF26-mediated ubiquitination as a potent modulator of IFN signalling (*Qin et al., 2014*). One model posits that RNF26 competes with RNF5 to extend K11- rather than K48-ubiquitin chains on the K150 residue of STING (*Qin et al., 2014*), thus suggesting the dynamic balance of ER-E3 ubiquitination governs STING's propensity for degradation, based on affinities of ubiquitin-binding proteasome subunits for different linkages. Domain mapping demonstrated RNF26 and STING interact through their TMDs (*Qin et al., 2014*), but further investigation is required to clarify the exact nature of this interaction and its impact on ubiquitination. We found RNF26 capable of modifying itself with K63-, K33-, and/or K29-Ub chains, but not K11- as shown previously for STING (*Qin et al., 2014*). This suggests distinct ubiquitination reactions occur in cis- and trans- by RNF26 and raises the potential involvement of different E2s and/or other co-factors in these processes.

## An RNF26 complex scales the STING-mediated IFN response

We identified four RNF26 interactors capable of scaling the IFN response downstream of STING activation. These modulators were not found previously by RNF26 proteomics, which only used the RING-containing, cytoplasmic C-terminus as bait (*Jongsma et al., 2016*). Together with RNF26, TMEM43, TMEM33, ENDOD1, and TMED1 form a membrane-bound complex that is capable of influencing STING signalling, most likely through TMD-based interactions and/or ubiquitination. Although these RNF26 HCIPs do not appear structurally similar or functionally orthologous, they share the ability to scale IFN signalling through the cGAS-STING pathway in an RNF26-dependent manner (*Figure 6E*), suggesting they act collectively through one or more complexes. How RNF26 and its HCIP accomplish this is not yet clear, as their individual functions are not yet fully appreciated. A comprehensive understanding of RNF26 complexes will require knockout cell lines to be made where functionality can be probed through rescue by expression constructs.

Loss of TMEM43 or TMEM33 increases the IFN response (*Figure 6*), consistent with roles as negative regulators of cGAS-STING signalling. As TMEM43 was required for RNF26 interaction with TMEM33 (*Figure 5E*), enhanced signalling through STING could be explained by the loss of TMEM33 from the RNF26 complex. TMEM33 is the human orthologue of *S. pombe* Tts1, an ER-

shaping protein that helps sustain high-curvature ER membranes (*Zhang et al., 2010*) and is involved in nuclear envelope remodelling during mitosis (*Zhang and Oliferenko, 2014*). Functional conservation in metazoans could indicate that local ER membrane curvature is an important determinant of STING activation. Alternatively, its role in organising the peripheral ER and as a reticulon binding protein could mean that TMEM33 influences localisation of RNF26 and STING throughout the ER/ INM network, which then determines signalling capability and regulation. TMEM43/LUMA localises in both the INM and the ER and interacts with proteins such as Lamin A/B and Emerin (*Bengtsson and Otto, 2008*). Mutations in TMEM43 are genetically linked to the heritable cardiomyopathy autosomal dominant arrhythmogenic right ventricular cardiomyopathy/dysplasia (ARVC/D, S358L) (*Merner et al., 2008*) and the autosomal recessive myopathy Emery-Dreifuss Muscular Dystrophy (EDMD, Q85K, I91V, [*Liang et al., 2011*]). Whether these rare conditions are in some way attributable to the suppression of STING activation by TMEM43 mutants is not known. TMEM43 has been linked previously to immune signalling and NF-kB through an interaction with CARMA3/ CARD10 and EGFR (*Jiang et al., 2017*), but not previously to STING.

Silencing TMED1 or ENDOD1 phenocopied the reduction of *IFIT1* levels upon cGAMP treatment that is observed with loss of RNF26, consistent with both HCIPs functioning to either permit or enhance STING signalling. Interestingly, depletion of either RNF26 and TMED1 resulted in a decrease in basal STING levels, despite the knockdown of TMED1 leading to an increase in RNF26 (*Figure 5E*). Although seemingly at odds, a model where RNF26 and STING adopt a defined stoichiometry may be one way to integrate these observations - one where the stability of STING may be tied to any changes in RNF26 levels. TMED1 is a member of the p24 family of trafficking proteins, and future research should establish whether vesicular trafficking from the ER alters STING and/or RNF26 levels. This is supported by recent findings showing the depletion of TOLLIP, an endocytic adaptor recruited by RNF26 to maintain endosomal positioning (*Jongsma et al., 2016*), also results in the loss of steady-state STING levels (*Pokatayev et al., 2020*). ENDOD1 is predicted to contain a non-specific endonuclease domain shown to have nuclease activity in vitro using an orthologous domain from *Paralichthys olivaceus* (Japanese flounder) (*Lyu et al., 2016*). Importantly, this study identified ENDOD1 among the genes that are important for innate immunity in fish, suggesting a role that is evolutionarily conserved. It is not yet clear from our study whether ENDOD1 and STING interact directly or what may be the implications of having endonuclease activity near a hub for immune signalling.

Our data reveal that RNF26 nucleates an immuno-regulatory complex, which was discovered through constructing the interaction landscape of the ER-resident E3s. Information within this landscape provides a resource to uncover E3 functions within various cellular processes. Understanding the mechanisms modulating abundance and activity of membrane-embedded proteins in the ER will help to determine if they represent tractable targets that may be leveraged for potential therapeutic benefit.

## Materials and methods

### Plasmids and transfections

All cDNAs encoding individual ER-E3s (*Supplementary file 1*, Table 9) were amplified and appended with restriction site-containing linkers by PCR, and subsequently subcloned into a pcDNA/5/FRT/TO vector (Invitrogen) containing a FLAG-HA (FH) tag in frame (N- or C-terminal) by restriction digest and ligation. Sequences for HCIPs (*Supplementary file 1*, Table 10) were obtained and processed similarly but subcloned instead into pcDNA3.1(-) vectors containing either an N- or C-terminal S-tag in frame as described previously (*Schulz et al., 2017*). All plasmids were transfected into recipient cell lines using Lipofectamine2000 (Thermo Fischer Scientific) according to manufacturer's guidelines.

### Cell culture and generation of stable cell lines

Flp-In T-REx human embryonic kidney 293 cells (referred to as Flp-In293) were originally obtained from their commercial source (Thermo Fischer Scientific). Flp-In293 cells were used to generate stable cell lines individually expressing selected E3s. Briefly, each FH-E3-pcDNA5/FRT/TO construct was co-transfected with the Flp recombinase vector pOG44 (3:1 ratio) as described above. Cell

pools stably recombining and expressing E3s were selected by resistance to Hygromycin B (100 μg/ ml, InvivoGen). All Flp-In293 cell lines were cultured in DMEM (Lonza, BE12-604F) supplemented with 10% (v/v) fetal bovine serum (FBS) and glutamine (2 mM). All cells were grown at 37°C and 5% $CO_2$. Mycoplasma was routinely tested for and all cell lines' negative status confirmed monthly.

## Antibodies and compounds

The following primary antibodies were used for detection by Western blot: anti-Hrd1 (Bethyl, #A302-946A), anti-SEL1L (Santa Cruz Biotechnology, #sc-48081), anti-UBE2J1 (Abcam, #ab39104), anti-OS-9 (kind gift from R. Kopito, Stanford), anti-Herp (Abcam, #ab150424,) anti-Derlin1 (kind gift from Y.Ye, NIH), anti-tubulin (Sigma, #T6074), anti-FLAG (Sigma, #F3165 and #F7425), anti-HA (Sigma, #H9658; Cell Signaling Technologies, #3724) anti-ubiquitin (Cell Signaling Technologies, #3933 and #3936), anti-RNF26 (Abcam, #ab57723), anti-S-tag (Thermo Scientific, #MA1-981), anti-TMEM43 (Abcam, #ab184164), anti-TMED1 (Abcam, #ab224411), anti-TMEM33 (Bethyl, #A305-597A-M), anti-ENDOD1 (Abcam, #ab121293), anti-AUP1 (Atlas Antibodies, #HPA007674), anti-UBXD8 (Proteintech, #16251–1-AP), anti-STING (Cell Signaling Technologies, #13647), anti-CD147 (Santa Cruz Biotechnology, #sc-25273). Anti-FAM8A1 has been reported previously (*Schulz et al., 2017*). The secondary antibodies used for western blot and IF include: goat anti-rabbit HRP (1:10,000, BioRad), goat anti-mouse HRP (1:10,000, Santa Cruz Biotechnology), donkey anti-goat HRP (1:10,000, Santa Cruz Biotechnology), goat anti-rabbit-Alexa 488 (1:400, Life Technologies), goat anti-mouse-Alexa 568 (1:400, Life Technologies). The following compounds were used in this study; MG132 (10 μM, Merck Millipore), Tunicamycin (500 ng/ml, Sigma), NMS-873 (10 μM, Sigma), cycloheximide (100 μg/ml, Abcam), N-ethylmaleamide (NEM, Acros Organics), DAPI (Sigma), doxy- cycline (DOX, Sigma), dithiothreitol (DTT, Sigma), iodoacetamide (IAA, Roche) and SubAB (*Paton et al., 2006*).

## siRNA transfections

For E3 knockdown, Flp-In293 cells seeded in 12-well plates were transfected with pooled siRNAs (100 nM, ON-TARGET SMARTpool, Dharmacon) using Dharmafect1 (Dharmacon) at a ratio of 1:4 according to the manufacturer's instructions. Cells were expanded 24 hr post-transfection and har- vested following another incubation of 24 hr. For knockdown of RNF26 and its associated interac- tors, Flp-In293 cell lines seeded in 12-well plates were transfected with individual siRNAs (50 nM, Sigma, *Supplementary file 1*, Table 11) using Lipofectamine RNAiMax (Thermo Fischer Scientific) at a ratio of 1:4 according to the manufacturer's instructions. Cells were expanded 24 hr post-transfec- tion and harvested following another incubation of 24 hr. RNF26 knockdown was confirmed by qRT- PCR (RNF26_F: GAATCCCCTCCTACCCCTGT, RNF26_R: GAGGAGAGCCCACAGCAAAT, Actin_F: GAGGCACTCTTCCAGCCTT, Actin_R: AAGGTAGTTTCGTGGATGCC), while interactor knockdowns were confirmed by western blot.

## Mass spectrometry and proteomic analysis

Each FH-E3-expressing Flp-In293 cell line was seeded in 15 cm plates and treated with DOX (1–1000 ng/ml, 18 hr). Cells were harvested at ~80% confluence, washed and subsequently resuspended in solubilisation lysis buffer (SLB: 150 mM NaCl, 50 mM Tris-HCl pH7.4, 5 mM EDTA) containing 1% Lauryl Maltose Neopentyl Glycol (LMNG, Anatrace) and supplemented with cOmplete protease inhibitor cocktail (Roche). Lysates were clarified by centrifugation (20,000 x g, 30 min.) and pre- cleared using CL-4B Sepharose beads (50 μL of 50:50 slurry, Pharmacia/GE). The resulting clarified whole cell lysate (WCL, 10 mg) was used as source material for immunoprecipitations with anti-FLAG agarose (M2, Sigma, A2220) for 2 hr. Immunoprecipitated complexes were washed twice with SLB (without detergent), twice with TBS, and eluted by 2 x SDS + 10% β-Mercaptoethanol. Eluates were reduced by DTT, alkylated by IAA, and subject to double chloroform-methanol precipitation. Precipi- tated proteins were subject to tryptic digest prior to purification using C18 Sep-Pak cartridges (Waters). Purified peptides were analysed by LC-MS/MS with a tandem mass spectrometer (Q Exac- tive HF, Thermo Fischer Scientific) with an EASY-Spray C18 LC Column (2 μm, 100 Å, 75 μm x 50 cm, Thermo Fischer Scientific) over a 63 min 2–35% acetonitrile gradient in 5% DMSO (v/v)/0.1% formic acid (v/v). The data were acquired with a resolution of 70,000 full-width half maximum at mass/ charge 400 with lock mass enabled (445.120025 m/z), Top 15 precursor ion selection, Dynamic

Exclusion of 27 s, and fragmentation performed in Higher-energy Collisional Dissociation mode with Normalized Collision Energy of 28. Samples were analysed twice to generate technical duplicates. Chromatogram alignment and peptide intensity were determined by Progenesis-QI (Nonlinear Dynamics). Peptides were identified by Mascot (version 2.5.1, Matrix Science) search against the SwissProt database (retrieved 26/11/2015). Each bait sample was assigned with more than 2000 protein IDs. Assignment of p-values to identified proteins was accomplished by adapting the comparative BSCG method described previously (*Keilhauer et al., 2015*). High-confidence interacting proteins (HCIPs) were defined by an ability to meet four criteria: 1) p-value<0.05, 2) positive fold-change, 3) identified by $\geq 2$ peptides, and 4) not classified as 'common' contaminants (*Supplementary file 1*, Table 3). SINQ analysis (*Trudgian et al., 2011*) was also carried out to identify interactors unique to one E3 only, which would not be assigned a p-value with the comparative analytical method. Apart from being unique to one sample, SINQ HCIPs were defined as: 1) having a SINQ score of $\geq 1 \times 10^{-7}$, 2) being identified by >1 spectral count, and 3) not being classified as a 'common' contaminant (*Supplementary file 1*, Table 3). The mass spectrometry proteomics data have been deposited to the ProteomeXchange Consortium via the PRIDE (*Perez-Riverol et al., 2019*) partner repository with the dataset identifier PXD019559 and 10.6019/PXD019559. Along with the Mascot search results, the Progenesis output, and BSCG calculation script and output are also included.

## Immunoprecipitation, SDS-PAGE and western blotting

Cells rinsed in phosphate-buffered saline (PBS) were mechanically lifted, harvested, and lysed in SLB + 1% LMNG or Triton X-100 (TX-100, Fischer Scientific), as described above. Lysates were clarified by centrifugation (17,000 x g, 30 min.) and pre-cleared using CL-4B Sepharose beads (50 µL of 50:50 slurry, Pharmacia/GE), with subsequent affinity- and immuno-purifications carried out using the resulting lysates. Beads were washed thrice in SLB and subsequently resuspended in 2 x Laemmli buffer + 20 mM DTT (10 min, 56°C) after the final wash, separated by SDS-PAGE and transferred to PVDF membrane for western blotting. Western blots were performed by incubating membranes in PBST blocking buffer (PBS + 1% Tween-20 supplemented with 5% non-fat dry milk), with subsequent primary and secondary antibody incubations in PBST + 5% non-fat dry milk. Secondary antibodies conjugated with horseradish peroxidase (HRP) were used to detect proteins bound to primary antibodies for enhanced chemiluminescence (ECL) with images captured either on X-ray film (FujiFilm, SuperRX) or by CCD camera (Chemidoc, BioRad) for quantification.

## Quantitative transcript analysis by NanoString

RNA from Flp-In293 cells was extracted using the RNeasy kit (Qiagen) in accordance with the manufacturer's instructions that included genomic DNA digestion with DNaseI (Qiagen). Isolated RNA from each sample (150 ng) was hybridized to a Reporter CodeSet and Capture ProbeSet (10 µL each) for a selected set of genes (*Supplementary file 1*, Table 6) by incubating in hybridization buffer (65°C, 18 hr) and loaded in the nCounter PrepStation according to manufacturer's instructions. Hybridized probe/target complexes were immobilized on the nCounter Cartridge and imaged in the nCounter MAX Digital Analyzer (high-resolution setting). Data were processed and analysed according to the manufacturer's guidelines (NanoString Technologies Inc). All experiments were performed in biological triplicate (n = 3).

## Radiolabelling and pulse-chase

Radiolabel pulse-chase assays of Flp-In293 cells stably expressing either FH-RNF26$_{WT}$ or FH-RNF26$_{Y432A}$ were carried out as previously described (*Christianson et al., 2008*; *Schulz et al., 2017*). Briefly, following DOX treatment (18 hr) cells were starved in DMEM (Lonza) lacking methionine (Met) and cysteine (Cys) + 10% dialysed FBS for 10 min, metabolically labelled by supplementing starvation medium with $^{35}$S-Met/Cys (EXPRE$^{35}$S$^{35}$S Protein Labelling Mix (PerkinElmer), 80 µCi/6 cm plate) for 10 min, rinsed thrice in PBS, and chased for indicated time points in DMEM supplemented with Met and Cys (50 mM each). Cells were lysed in SLB containing 1% TX-100, and the detergent-soluble, post-nuclear lysates pre-cleared using CL-4B Sepharose beads followed by immunoprecipitation with anti-HA-antibody (12CA5) and Protein G agarose (Roche). Bead-bound

radiolabelled substrates were resuspended in 2 x Laemmli buffer (+20 mM DTT), separated by SDS-PAGE and imaged using a phosphoimager (BioRad).

## cGAMP transfection for *IFIT1* qRT-PCR

Flp-In293 cells seeded in 24-wells were stimulated by transfecting 5 µg/ml cGAMP using Lipofect-amine 2000 at a ratio of 1.25:1. Cells were harvested 6 hr post-transfection and the extracted RNA (RNeasy, Qiagen) reverse transcribed to produce cDNA (QuantiTect, Qiagen) according to manufacturer's instructions. Taqman probes targeting human *GAPDH* (Hs02758991_g1) and *IFIT1* (Hs03027069_s1) were purchased from Life Technologies. qRT-PCR data were collected on a StepO-nePlus Thermal Cycler (Thermo Fischer Scientific) and analysed by the $\Delta\Delta C_t$ method, normalising *IFIT1* levels to GAPDH. Averages and S.E.M. were determined from at least three independent experiments (n = 3).

## *XBP1* splicing assay

ER stress was induced in Flp-In293 cells by treatment with 5 mM DTT. Cells were harvested after 2 hr and subject to RNA extraction and cDNA synthesis as described above. Primer sequences and PCR conditions are from *Lin et al., 2007*. PCR products were separated on a 2.5% (w/v) agarose gel.

## Immuno- and affinity purification of ubiquitinated proteins

FH-RNF26$_{WT}$ and FH-RNF26$_{Y432A}$ -expressing Flp-In293 cells were induced with DOX (18 hr) and where indicated, samples were additionally treated with 10 µM MG132 for 2 hr. Cell pellets were lysed in TUBE lysis buffer (20 mM sodium phosphate pH 7.5, 1% NP-40 (v/v), 2 mM EDTA, supplemented with cOmplete protease inhibitor cocktail (Roche), PhosSTOP (Roche), NEM (50 mM), and DTT (1 mM)). Lysates were centrifuged (as above) and the detergent-soluble fraction subsequently incubated overnight with 15 µL magnetic GST resin (Thermo Fischer Scientific) conjugated to 50 µg 1x UBA-His$_6$ binder (*Hjerpe et al., 2009*; *Hrdinka et al., 2016*). Bound resin was washed thrice with TUBE lysis buffer and each sample split equally to accommodate control/untreated or USP21 deubiquitinase treatment.

## Deubiquitination assays

Bead-bound material from UBA-His$_6$ binder pulldowns (above) was resuspended in deubiquitinating-buffer (50 mM HEPES pH 7.5, 100 mM NaCl, 2 mM DTT, 1 mM MnCl$_2$, 0.01% Brij-35) without or with 0.5 µM USP21 (Ubiquigent). Samples were incubated (1 hr, 30°C) in a thermoshaker (VWR, 750 rpm) and subsequently denatured by incubation (65°C, 20 min) with 2 x Laemmli buffer + DTT (20 mM) followed by separation on SDS-PAGE. For UbiCREST analysis (*Hospenthal et al., 2015*), FH-RNF26$_{WT}$ from stably expressing Flp-In293 cells (4 mg) was immunoprecipitated by anti-FLAG agarose, washed, divided and individually incubated with the panel of recombinant DUBs, according to the manufacturer's instructions (Boston Biochem).

## In vitro ubiquitination assay

FH-RNF26$_{WT}$ and FH-RNF26$_{Y432A}$ -expressing Flp-In293 cells were treated and lysed in TUBE lysis buffer as described above. Resulting supernatants were incubated with anti-FLAG M2 magnetic beads (Sigma, 3 hr at 4°C). Beads were washed thrice with TUBE lysis buffer, dividing samples in half prior to the last wash and then washed once with in vitro ubiquitination (IVU) buffer (50 mM Tris-HCl, pH 7.5, 2.5 mM MgCl$_2$). Assays were carried out by resuspending beads in IVU buffer supplemented with; ubiquitin (10 µM, Boston Biochem), E1 enzyme (150 nM, Enzo), UbcH5a (1 µM, Enzo), and DTT (0.5 mM), with or without ATP (4 mM, pH 8.0, Sigma) and incubated in a thermoshaker (15 min, 30°C, 750 rpm). Samples were denatured with 2 x Laemmli buffer + 20 mM DTT (65°C, 20 min).

## Velocity sedimentation

Velocity sedimentation was carried out as previously described (*Schulz et al., 2017*). Briefly, DOX-induced Flp-In293 cells (18 hr) were mechanically harvested and lysed in SLB containing 1% LMNG (as described above). Post-nuclear, pre-cleared WCLs (1 mg total) were layered onto either a continuous sucrose gradient (10–40% or 5–30%) prepared using a Gradient Master 108 (BioComp).

Sucrose was dissolved in a physiological salt solution (150 mM NaCl, 50 mM Tris-HCl pH 7.4, 5 mM EDTA, 1 mM PMSF) + 1% LMNG. Samples were centrifuged in an SW.41 rotor (OptimaTM L-100 XP, Beckman Coulter, Brea, CA) at 39,000 rpm for 16 hr at 4°C. Thirteen fractions (940 μL each) were collected manually and proteins precipitated by addition of 190 μL 50% (v/v) trichloroacetic acid (TCA). Following acetone washes, precipitated proteins were resuspended in 2 x Laemmli buffer + 20 mM DTT (10 min, 56°C) and separated by SDS-PAGE. If necessary, samples were neutralised with 1 M Tris-HCl (pH 9). Gel filtration standards (Gel Filtration Markers Kit, MWGF1000, Sigma Aldrich) were separated on similar gradients to estimate protein complex size and included: alcohol dehydrogenase (150 kDa), β-amylase (200 kDa), apoferritin (443 kDa) and thyroglobulin (663 kDa). Standards were processed as above and detected by Coomassie staining.

## Immunofluorescence and microscopy

For detection of FH-E3s, Flp-In293 cells were seeded onto 13 mm poly-L-lysine coated cover slips and induced with DOX (18 hr). Cells were fixed with 4% paraformaldehyde (PFA, 20 min at room temperature (RT)), permeabilised with PBS containing 0.2% TX-100 for 5 min at RT and blocked with PBS containing 0.2% PBG (fish skin gelatin) for 30 min. Coverslips were incubated with 1° antibodies diluted in 0.2% PBG (1 hr, RT), rinsed twice in PBS and incubated with fluorescent 2° antibodies (0.2% PBG in PBS, 1 hr, RT). Coverslips were incubated with DAPI (5 μg/mL, 10 min, RT) and mounted using ProLong Gold antifade reagent (Life Technologies). All images were captured on a Zeiss LSM710 confocal microscope and processed in Image J (NIH) and Photoshop (Adobe).

## Statistical analysis

Statistical significance within NanoString data was determined using multiple t-tests (Holm-Sidak method, $\alpha$ = 0.05) that compared fold change in E3 transcripts from untreated and Tunicamycin-treated (Tm, 500 ng/ml, 8 hr) or SubAB-treated (10 ng/ml, 8 hr) samples. All other data (e.g. qPCR, protein quantification) were analysed using a two-tailed paired t-test comparing non-targeting control siRNA (siNTC) to each siRNA target. All statistical analyses were carried out and plotted using GraphPad Prism (Version 7.0). Detailed statistical information is available in *Supplementary file 1*, Table 12.

## Bioinformatic analysis

Primary amino acid sequences for all E3s and HCIPs were obtained from UniProt (http://www.uniprot.org/), with common motifs annotated using Pfam (http://pfam.xfam.org/) (*Finn et al., 2016*), TMDs predicted by TOPCONS (http://topcons.net/) (*Tsirigos et al., 2015*) and N- linked glycosylation sites predicted by NetNGlyc 1.0 (http://www.cbs.dtu.dk/services/NetNGlyc/). E3 interactions were compared against those previously reported in BioGRID 3.5 (https://thebiogrid.org/) and BioPlex 3.0 (https://bioplex.hms.harvard.edu/) databases.

## Acknowledgements

We are grateful to Dr. Norbert Volkmar and Dr. Dönem Avci for critical discussions. We also thank Dr. Jan Rehwinkel for technical assistance. EF was supported by a fellowship from the Medical Research Council. PDC was supported by an EPSRC grant (nr EP/N034295/1) and by the Chinese Academy of Medical Sciences (CAMS) Innovation Fund for Medical Science (CIFMS), China (grant number: 2018-I2M-2–002), awarded to BMK. The MG-H lab was supported by the Ludwig Institute for Cancer Research and a Wellcome Trust Fellowship (102894/Z/13/Z). JCC was supported by a grant from the Medical Research Council (MR/L001209/1) and by the Ludwig Institute for Cancer Research.

## Additional information

### Funding

| Funder | Grant reference number | Author |
| --- | --- | --- |
| Medical Research Council | Graduate Student Fellowship | Emma J Fenech |

| Engineering and Physical Sciences Research Council | EP/N034295/1 | Philip D Charles |
|---|---|---|
| Chinese Academy of Medical Sciences | 2018-I2M-2-002 | Philip D Charles Benedikt M Kessler |
| Medical Research Council | MR/L001209/1 | John C Christianson |
| Wellcome | 102894/Z/13/Z | Mads Gyrd-Hansen |
| Ludwig Institute for Cancer Research | | John C Christianson |
| Ludwig Institute for Cancer Research | | Mads Gyrd-Hansen |

The funders had no role in study design, data collection and interpretation, or the decision to submit the work for publication.

## Author contributions

Emma J Fenech, Data curation, Formal analysis, Validation, Investigation, Visualization, Methodology, Writing - original draft, Writing - review and editing; Federica Lari, Data curation, Formal analysis, Validation, Investigation, Visualization, Methodology; Philip D Charles, Data curation, Formal analysis, Investigation, Visualization, Methodology, Writing - original draft, Writing - review and editing; Roman Fischer, Resources, Supervision; Marie Laétitia-Thézénas, Formal analysis; Katrin Bagola, Investigation, Methodology; Adrienne W Paton, James C Paton, Resources, Writing - review and editing; Mads Gyrd-Hansen, Resources, Supervision, Funding acquisition, Writing - review and editing; Benedikt M Kessler, Resources, Supervision, Funding acquisition, Methodology, Writing - original draft, Writing - review and editing; John C Christianson, Conceptualization, Resources, Data curation, Formal analysis, Supervision, Funding acquisition, Validation, Investigation, Visualization, Methodology, Writing - original draft, Project administration, Writing - review and editing

## Author ORCIDs

Emma J Fenech (iD) https://orcid.org/0000-0003-4414-3233
Philip D Charles (iD) http://orcid.org/0000-0001-5278-5354
Roman Fischer (iD) http://orcid.org/0000-0002-9715-5951
Mads Gyrd-Hansen (iD) https://orcid.org/0000-0001-5641-5019
Benedikt M Kessler (iD) http://orcid.org/0000-0002-8160-2446
John C Christianson (iD) https://orcid.org/0000-0002-0474-1207

## Decision letter and Author response

Decision letter https://doi.org/10.7554/eLife.57306.sa1
Author response https://doi.org/10.7554/eLife.57306.sa2

# Additional files

## Supplementary files

• Supplementary file 1. Data Tables 1-12: ER-E3 LC-MS/MS (1-8) and cDNA/siRNA sequence (9-12) supporting information. Supplemental Table 1- ER-resident E3 features and positioning of FLAG-HA epitopes. Supplemental Table 2- Unique protein signatures of ER-resident E3 pulldowns identified by LC-MS/MS and BSCG analysis. Supplemental Table 3- Common contaminants removed from BSCG and SINQ analyses. Supplemental Table 4- Unique ER-resident E3 interactors identified by SINQ analysis of LC-MS/MS dataset. Supplemental Table 5- High-confidence candidate interacting proteins (HCIPs) of ER-resident E3s. Supplemental Table 6- NanoString quantitative transcriptomic codeset of ER-resident E3s. Supplemental Table 7- Targets of UPR transcription factors among ER-resident E3 HCIPs. Supplemental Table 8- BSCG analysis of $RNF26_{Y432A}$ compared to $RNF26_{WT}$. Supplemental Table 9- ORF sources for ER-resident E3. Supplemental Table 10- ORF sources for HCIPs Supplemental Table 11- siRNA sequences Supplemental Table 12. Statistical analyses.

• Transparent reporting form

### Data availability

All data generated or analysed during this study are included in the manuscript and supporting files.

The following dataset was generated:

| Author(s) | Year | Dataset title | Dataset URL | Database and Identifier |
|---|---|---|---|---|
| Christianson JC | 2020 | Interaction mapping of endoplasmic reticulum ubiquitin ligases identifies modulators of innate immune signalling | https://www.ebi.ac.uk/pride/archive/projects/PXD019559 | PRIDE, PXD019559 |

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
