## [Decision Letter]

**Acceptance summary:**

A number of ubiquitin ligases contain transmembrane segments and are associated with the endoplasmic reticulum. This paper provides what may be the most comprehensive analysis of interaction partners for ubiquitin ligases associated with the endoplasmic reticulum. The authors follow up on one interaction linking RNF26 to the STING pathway, and identify a multiprotein complex important for RNF26 function. This work will provide a valuable resource for the field.

**Decision letter after peer review:**

Thank you for submitting your article "Interaction mapping of endoplasmic reticulum ubiquitin ligases identifies modulators of innate immune signalling" for consideration by *eLife*. Your article has been reviewed by three peer reviewers and the evaluation has been overseen by David Ron as the Senior Editor. The reviewers have opted to remain anonymous.

The reviewers have discussed the reviews with one another and the Reviewing Editor has drafted this decision to help you prepare a revised submission.

Summary:

This paper examines the interactomes of 21 ubiquitin ligases that are known to be associated with the endoplasmic reticulum (ER). Numerous candidate interacting proteins are identified. The authors follow up one complex in detail, which is composed of RNF26 and a 4 protein membrane complex – TMEM43, ENDOD1, TMEM33 and TMED1. They link this complex to the cGAS-STING pathway suggesting a signaling function for the ER in innate immunity.

Essential revisions:

The reviewer's felt that the interactome data was well done and would be a valuable resource for the community. However, it is also the case that there is not a lot of validation of interactions.

There are other issues related to the functional studies that should be address, with data if available already but definitely within the text:

1) The data in Figure 4 demonstrates that overexpressed RNF26 is unstable and rapidly degraded by the proteasome. These data are clear and convincing, but it remains unclear if the endogenous protein is similarly unstable. Is the overexpressed protein unstable because it lacks a key oligomeric binding partner or because it is being expressed in a non-native cell type?

2) Similar to comment 1, it should be pointed out that while the sucrose gradient experiments are suggestive of distinct complexes, it is possible that the overexpression of RNF26 alters the complex stoichiometry.

In many ways, the paper is more like a resource. One alternative might be to have the paper as a Tools and Resources article. This may be more appropriate given some of the experiments that are missing from the analysis (see below).

Revisions expected in follow-up work:

The work would benefit from a future analysis of cells deleted for RNF26. In addition, there are several questions related to the mechanism of STING regulation that deserve future attention. This includes a more definitive analysis of the role of the TMEM43/ENDOD1/TMEM33/TMED1 complex in STING regulation and a better understanding of exactly how RNF26 ubiquitylates STING. Although there was a prior paper on the regulation of STING by RNF26, this paper could be classified as quite preliminary and lacking in mechanistic detail. One possibility that comes to mind might be to directly measure sting ubiquitylation in the context of distinct deletions of the RNF26 complex members. It would also be important to further validate interactions within this complex at endogenous protein levels.

---

## [Author Response]

Essential revisions:The reviewer's felt that the interactome data was well done and would be a valuable resource for the community. However, it is also the case that there is not a lot of validation of interactions.There are other issues related to the functional studies that should be address, with data if available already but definitely within the text:

We thank the editors and reviewers for their insightful perspectives and comments on our manuscript, and also for their recommendation to re-submit this manuscript as a ‘Tools and Resources’ article. We fully agree with the reviewers that our work serves more as a resource, and therefore accept this recommendation. Where appropriate, we have addressed any issues raised within the text of the updated manuscript and indicate them accordingly.

1) The data in Figure 4 demonstrates that overexpressed RNF26 is unstable and rapidly degraded by the proteasome. These data are clear and convincing, but it remains unclear if the endogenous protein is similarly unstable. Is the overexpressed protein unstable because it lacks a key oligomeric binding partner or because it is being expressed in a non-native cell type?2) Similar to comment 1, it should be pointed out that while the sucrose gradient experiments are suggestive of distinct complexes, it is possible that the overexpression of RNF26 alters the complex stoichiometry.

The reviewers highlight an important issue regarding the endogenous form of RNF26 and whether it would behave similarly to what we observe for the doxycycline-induced, epitope-tagged expressed RNF26 form in terms of (1) intrinsic stability and (2) protein-protein interactions/complex formation. We have tried to address these points in the Results (subsection “RNF26 is unstable and degraded by the ubiquitin-proteasome system (UPS)”) and Discussion (subsection “RNF26 is an intrinsically unstable ER-E3) within the sections on RNF26 instability. Briefly, a number of lines of evidence suggest the difficulty to robustly detect endogenous RNF26 is due to intrinsic instability, as we observe with the DOX-induced variant. RNF26 mRNA levels are comparable to other E3s (Figure 4D) yet detection of endogenous forms is only feasible when stabilised as a putative dimer by a ubiquitination-defective mutant (Figure 5E). Other groups have reported similar difficulties with endogenous RNF26 detection in a different cell type (Jongsma et al., 2016, Ilana Berlin (Leiden) – personal communication). We know less about RNF26 stoichiometry or whether any of the identified HCIPs are limiting for complex formation, where overexpression would lead to formation of non-native interactions or unbalanced stoichiometry. This has now been mentioned (subsection “Discovery and identification of RNF26 interactors”, second paragraph). Tagged RNF26 overexpression could impact complex stoichiometry, but we did observe that its HCIPs appeared to be turned over at a rate much slower than RNF26 (Figure 5F). While not speaking directly to changes in stoichiometry, it does suggest that the stability of its HCIPs are not markedly influenced by RNF26 turnover.